# ATAD5 promotes replication restart by regulating RAD51 and PCNA in response to replication stress

Su Hyung Park[1], Nalae Kang[1], Eunho Song [2,3], Minwoo Wie[4], Eun A. Lee[1], Sunyoung Hwang[1], Deokjae Lee[4,6], Jae Sun Ra[1], In Bae Park[1], Jieun Park[1], Sukhyun Kang[1], Jun Hong Park [1], Sungchul Hohng [2,3,5], Kyoo-young Lee[1]* & Kyungjae Myung[1,4]*

Maintaining stability of replication forks is important for genomic integrity. However, it is not clear how replisome proteins contribute to fork stability under replication stress. Here, we report that ATAD5, a PCNA unloader, plays multiple functions at stalled forks including promoting its restart. ATAD5 depletion increases genomic instability upon hydroxyurea treatment in cultured cells and mice. ATAD5 recruits RAD51 to stalled forks in an ATR kinase-dependent manner by hydroxyurea-enhanced protein-protein interactions and timely removes PCNA from stalled forks for RAD51 recruitment. Consistent with the role of RAD51 in fork regression, ATAD5 depletion inhibits slowdown of fork progression and native 5-bromo-2′-deoxyuridine signal induced by hydroxyurea. Single-molecule FRET showed that PCNA itself acts as a mechanical barrier to fork regression. Consequently, DNA breaks required for fork restart are reduced by ATAD5 depletion. Collectively, our results suggest an important role of ATAD5 in maintaining genome integrity during replication stress.

[1] Center for Genomic Integrity, Institute for Basic Science, Ulsan, Korea. [2] Interdisciplinary Graduate Program in Biophysics and Chemical Biology, Seoul National University, Seoul 08826, Republic of Korea. [3] Institute of Applied Physics, Seoul National University, Seoul 08826, Republic of Korea. [4] Department of Biological Sciences, School of Life Sciences, Ulsan National Institute of Science and Technology, Ulsan, Korea. [5] Department of Physics and Astronomy, Seoul National University, Seoul 08826, Republic of Korea. [6] Present address: Medytox Inc. 114, Yeongtong-gu, Suwon-si, Gyeonggi-do, Korea. *email: klee2910@ibs.re.kr; kmyung@ibs.re.kr

Maintaining genomic integrity is essential for cell survival and the accurate delivery of genetic information during cell division. Accumulating evidence suggests that replication stress is one of the major contributors to genomic instability in cancer cells[1]. Replication stress slows or stalls the progression of the replication fork, and if not properly resolved, the fork with exposed single-stranded (ss) DNA collapses. One way to process stalled forks is the reannealing of template DNA and subsequent nascent DNA to generate a four-way junction structure, which is referred to as fork regression[2]. Protecting and restarting stalled/regressed replication forks are two key processes to ensure faithful DNA replication under replication stress. Stalled/regressed replication forks are simply restored by fork reversal enzymes under mild replication stress, while prolonged replication stress likely produces aberrant fork structures that are restored by a mechanism that is yet to be defined in detail[3].

Many homologous recombination (HR) proteins have roles at stalled/collapsed replication forks besides their well-established roles in double-strand break (DSB) repair. They work co-operatively to prevent the excessive nucleolytic degradation of nascent DNA strands and to assist the regression and/or pro-cessing of stalled forks, thereby contributing to fork restart[4,5]. Among HR proteins, the RAD51 recombinase is essential for fork stability and fork regression[6,7]. RAD51 filament formation at the stalled forks is important for fork stability and regression. Only fork stability by RAD51 depends on BRCA2[5,8]. However, the molecular mechanisms by which RAD51 is recruited to or accumulated at stalled forks and which replisome protein(s) it communicates with to coordinate this process are not clear. Besides RAD51 and BRCA2, HR proteins involved in DSB resection or Holliday junction resolution are involved in fork processing under replication stress[4,9]. The structure-specific endonuclease MUS81-EME2 has been reported to be respon-sible for the cleavage and restart of stalled forks[10].

ATAD5 is important for maintaining genomic stability in eukaryotic organisms from yeast to humans[11,12] and the impor-tance of this function is underscored by the fact that *Atad5* heterozygote mutant mice develop tumors[13]. Additionally, somatic mutations of *ATAD5* have been found in patients with several types of cancer and a genome-wide analysis indicated that the *ATAD5* locus confers enhanced susceptibility to endometrial, breast, and ovarian cancers[13–15]. These observations suggest that ATAD5 functions as a tumor suppressor. ATAD5 forms an alternative pentameric replication factor C (RFC)-like complex (RLC) with the core subunits RFC2–5. We previously reported that ATAD5-RLC regulates the functions of the eukaryotic DNA polymerase processivity factor proliferating cell nuclear antigen (PCNA) by unloading the ring-shaped PCNA homotrimer from DNA upon its successful replication during the S phase of the cell cycle[16,17]. Additionally, ATAD5-RLC restricts the error-prone damage bypass pathway by recruiting the ubiquitin-specific protease 1 (USP1)/USP1-associated factor (UAF1)-deubiquiti-nating enzyme complex to reverse PCNA mono-ubiquitination, which is a modification required for DNA lesion bypass. It is still unclear which of the PCNA-regulating functions of ATAD5-RLC are important for its role as a tumor suppressor.

ATAD5-depleted cells show characteristic features of replica-tion stress such as a slow replication rate[17] and it has been sug-gested that the loss of PCNA-regulating activity of ATAD5 might be the cause of this phenotype. We hypothesized that there is a mechanism of ATAD5 in counteracting replication stress. We find that ATAD5-RLC plays important roles in restarting stalled forks under replication stress. ATAD5-RLC promotes RAD51 recruitment to stalled forks by direct protein–protein interaction. In addition, we report that PCNA unloading by ATAD5-RLC is a prerequisite for efficient RAD51 recruitment. Our data suggest

that a series of processes starting with RAD51 recruitment and leading to fork regression, breakage, and eventual fork restart are regulated by ATAD5. The way of ATAD5 maintaining genome stability, therefore, extends beyond its roles in PCNA unloading and deubiquitination.

## Results

**ATAD5 is important for restarting stalled replication forks.** We first attempted to assess whether ATAD5 plays a role in fork stability under replication stress using two different methods. Since ATAD5 depletion affects the cell cycle and the DNA replication rate (Fig. 1b, bottom panel and ref. [17]), we have established a new S-phase synchronization procedure called the Noco-APH condition combined with a short small interfering RNA (siRNA) treatment to minimize the cellular effects of ATAD5 depletion before exogenous replication stress is applied (Fig. 1a). Under these conditions, 50–70% of cells progressed to the S phase without DNA damage and checkpoint activation after being released from cell cycle arrest at the G1/S boundary, and subsequently re-entered the next G1 phase (Supplementary Fig. 1A–C). ATAD5 expression was reduced by the short siRNA treatment and consequently PCNA was accumulated on the chromatin (Supplementary Fig. 1D). More importantly, a flow cytometry analysis of 5-ethynyl-2′-deoxyuridine (EdU) incor-poration showed that the replication rate was comparable between the control and ATAD5-depleted cells under the Noco-APH condition (Fig. 1b, upper panel). To induce replication stress, cells were released from cell cycle arrest and treated with hydroxyurea (HU), which depletes cellular dNTP levels. Alter-natively, we have established an auxin-inducible degron (AID) cell line to rapidly deplete endogenous ATAD5 protein (Supple-mentary Fig. 1E). AID-tagged ATAD5 (ATAD5$^{AID}$) was degra-ded by auxin treatment, which was also confirmed by PCNA accumulated on the chromatin (Supplementary Fig. 1F).

We investigated whether ATAD5 depletion affects fork restart under replication stress using a DNA combing assay with two different protein depletion methods described above. The DNA combing assay involves the consecutive labeling of replicating DNA with chloro-deoxyuridine (Cl-dU) and iodo-deoxyuridine (I-dU), followed by detection of the labeled deoxyuridines on DNA spread using antibodies conjugated with different fluor-ophores. As previously reported, HU treatment after the first Cl-dU labeling reduced the detection of the second I-dU labeling due to failed fork restart (Fig. 1c–e). ATAD5 depletion by siRNA or AID further reduced fork restart, suggesting that ATAD5 facilitates restarting stalled forks under replication stress. HU-induced new origin firing was not significantly affected by ATAD5 depletion (Supplementary Fig. 1G, H).

Under replication stress, replication fork is actively slowed down and undergoes fork regression, a process that reanneals of nascent DNA to generate a four-way junction structure[7]. This process is considered to stabilize the stalled forks[2]. We investigated effects of ATAD5 depletion on replication fork speed under replication stress. The minimum dose of HU treatment induced a significant slowing of replication fork speed, but this was not observed in ATAD5-depleted cells (Fig. 1f). The native 5-bromo-2′-deoxyuridine (BrdU) assay can be used to detect single-stranded regions in DNA structure, including regressed fork. As previously reported, HU treatment increased the native BrdU signal, while ATAD5 depletion reduced the signal (Fig. 1g–i and Supplementary Fig. 1I). In addition, the reduced BrdU signal in ATAD5-depleted cells was not recovered by treatment with the MRE11 inhibitor mirin (Fig. 1j), thus excluding a possibility of nascent DNA degradation by the exonuclease activity of MRE11[4]. Taken together, these data

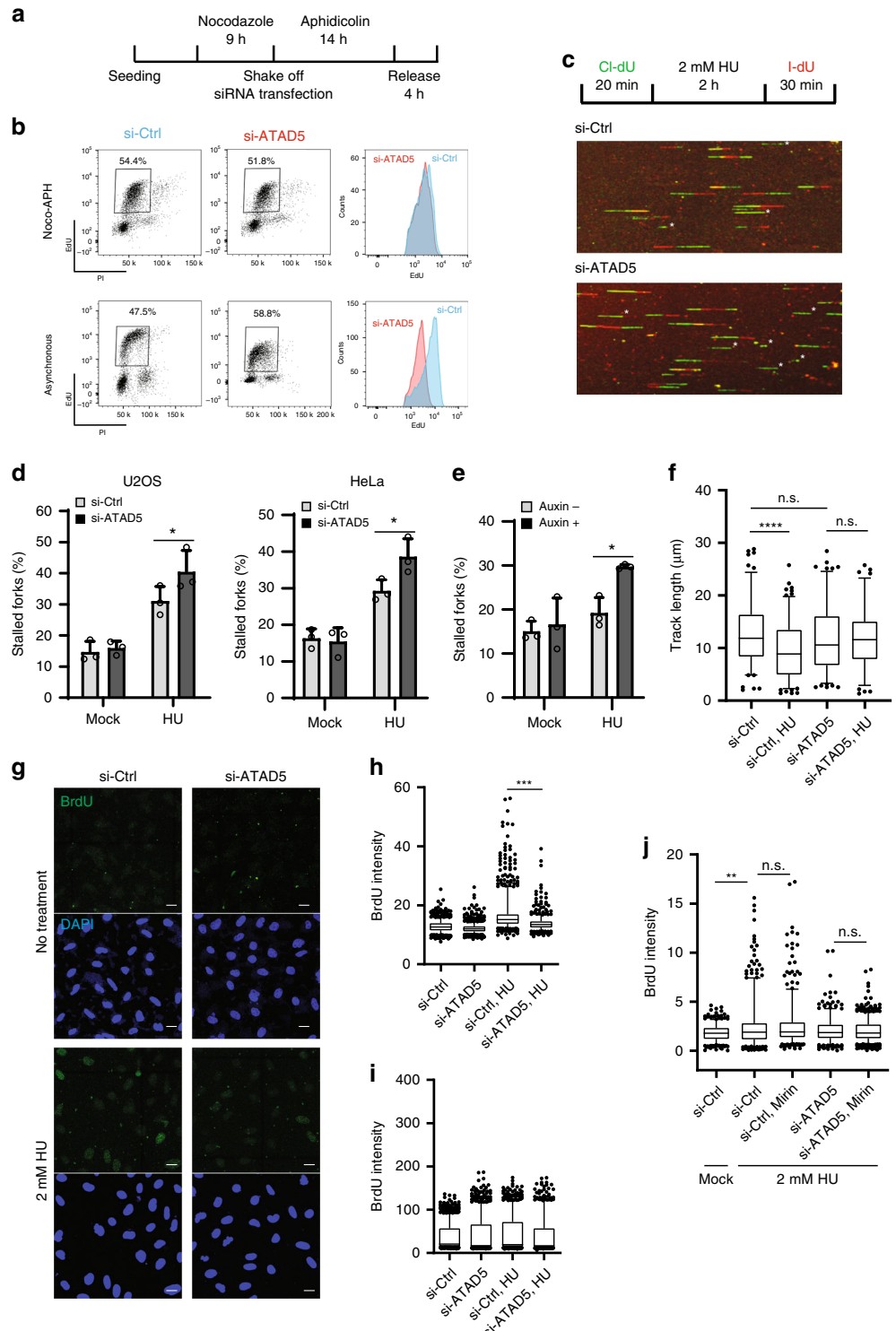

suggest that ATAD5 might be involved in fork regression under replication stress.

**ATAD5 promotes RAD51 recruitment to stalled forks**. RAD51 plays critical roles in fork regression and fork stability under replication stress[7]. Since both fork restart and possibly fork regression were reduced in ATAD5-depleted cells, we examined the effects of ATAD5 depletion on HU-induced RAD51 foci formation. As reported, HU treatment increased the chromatin-bound RAD51 signal (Fig. 2a, b). However, the HU-induced

increase in the chromatin-bound RAD51 signal was significantly reduced in ATAD5-depleted cells. In addition, the isolation of proteins on nascent DNA (iPOND) assay, which detects proteins at/near the replication fork based on EdU labeling and click chemistry, showed that the HU-induced recruitment of RAD51 was diminished in cells where ATAD5 was depleted by siRNA or AID (Fig. 2c and Supplementary Fig. 2A). These results suggest that ATAD5 is required for RAD51 recruitment to stalled forks. Next, we investigated whether ATAD5 and RAD51 show epistasis for fork restart (Fig. 2d). RAD51 depletion reduced fork restart, as reported[6], and simultaneous depletion of ATAD5 and RAD51 did

**Fig. 1 ATAD5 promotes replication fork restart at stalled replication forks. a** The scheme for cell cycle arrest (Noco-APH condition). U2OS cells were arrested at the G1/S boundary and then released from arrest in normal media for 4 h. Human *ATAD5* small interfering (si) RNA was transfected when cells were re-seeded after shaking-off. **b** U2OS cells released from arrest for 4 h were collected for cell cycle analysis. Asynchronous cells were transfected with *ATAD5* siRNA for 48 h before cell collection. Under both conditions, cells were pulse-labeled with 5-ethynyl-2′-deoxyuridine (EdU) for 30 min before cell collection. **c, d** U2OS or HeLa cells were transfected with *ATAD5* siRNA under the Noco-APH condition. **e** U2OS cells expressing ATAD5$^{AID}$ were pre-treated with auxin. **c–e** After depletion of ATAD5, cells were analyzed using a DNA combing assay. **c** Representative images of replication tract. Each asterisk (*) indicates a stalled replication fork. **d, e** The percentages of stalled forks are displayed. Error bars represent standard deviation of the mean ($n =$ 3). Statistical analysis: $t$ test; *$p < 0.05$. **f** U2OS cells were transfected with *ATAD5* siRNA under the Noco-APH condition. Then, cells were subjected for a DNA combing assay. The length of l-dU-labeled DNA track was measured. **g–j** U2OS cells transfected with *ATAD5* siRNA under the Noco-APH condition were labeled with 5-bromo-2′-deoxyuridine (BrdU) for 10 min and incubated with 2 mM HU for 4 h before fixation. **g, h, j** Single-stranded DNA with BrdU exposed was visualized by staining with an anti-BrdU antibody under native non-denaturing conditions. **i** Total BrdU in DNA was detected under denaturing conditions as a control. **g** Representative images of anti-BrdU antibody staining. Scale bar: 20 μm. **h, i** The intensity of BrdU staining was quantified from ~200 cells and plotted. Three independent experiments were performed and one representative result was displayed. **j** 50 μM mirin was simultaneously administered with 2 mM HU for 4 h before fixation. **f, h, i, j** Boxes indicate median and interquartile ranges and whiskers indicate the 5th–95th percentile. Statistical analysis: Mann–Whitney $U$ test; ****$p < 0.0001$, ***$p < 0.001$, **$p < 0.01$, n.s., not significant.

not show additional effects, suggesting that ATAD5 and RAD51 work in the same pathway for fork restart. The less effects in fork restart by ATAD5 depletion compared to RAD51 depletion suggest that there might be an additional pathway for RAD51 regulation.

ATAD5 has two PCNA-regulating functions, namely PCNA deubiquitination and PCNA unloading, which depend on the UAF1 interaction domain and the ATPase domain of ATAD5, respectively[16,17]. We investigated whether defects in either of these functions could be a cause of the reduction in RAD51 recruitment to stalled forks in ATAD5-depleted cells. We performed the quantitative in situ analysis of protein interactions at DNA replication forks (SIRF) assay, which examines the association of a protein with EdU-labeled nascent DNA at the single-cell level[18]. The association of RAD51 with nascent DNA was increased by HU treatment, which was reduced in ATAD5-depleted cells (Fig. 2e and Supplementary Fig. 2B, C). The siRNA-resistant wild-type ATAD5 completely restored the HU-induced association of RAD51 with nascent DNA, while the UAF1 interaction-defective and ATPase-defective mutants of ATAD5 did not (Fig. 2e). We also examined effects of PCNA-regulating functions of ATAD5 on HU-induced chromatin-bound RAD51 signal and HU-induced deceleration of replication fork progression (Fig. 2f, g). We used U2OS-TetOn cell lines expressing the wild-type or mutant ATAD5 proteins in a doxycycline-inducible manner in a cell in which native ATAD5 expression was repressed by siRNA. Consistent with previous reports, UAF1 interaction-defective and ATPase-defective mutants of ATAD5 displayed defects in PCNA deubiquitination and PCNA unloading, respectively (Supplementary Fig. 2D). The siRNA-resistant wild-type ATAD5 restored the staining intensity of chromatin-bound RAD51 upon HU treatment, while the ATAD5 ATPase mutant did not (Fig. 2f). Meanwhile, the UAF1 interaction-defective mutant ATAD5 partially restored the chromatin-bound staining intensity upon HU treatment (Fig. 2f). Consistently, the wild-type ATAD5 restored HU-induced deceleration of replication fork progression, whereas both mutants failed to complement it (Fig. 2g). These results suggest that both the PCNA unloading activity and the interaction with UAF1 are required for ATAD5 to efficiently recruit RAD51 to stalled forks.

Besides effects on RAD51 recruitment, PCNA itself accumulated on the lagging strand at the replication forks, which might mechanically inhibit proper fork regression in ATAD5-depleted cells. We tested the possibility by using a single-molecule FRET (fluorescence resonance energy transfer) experiment. We prepared a model replication fork labeled with Cy3 and Cy5 (Supplementary Fig. 3A) so that fork reversal dynamics can be studied using FRET[19,20]. Immobilized DNA forks on Quartz slide

with the end blocked with anti-digoxigenin to prevent fall-off of loaded Alexa488-labeled PCNA were incubated with WRN helicase to promote regression of DNA forks. FRET jumps indicate the arrival of the branch point of the four-way junction at the labeling site of the FRET probe, and the simultaneous disappearance of the Cy3 and Cy5 signals were observed as the completion of fork regression (Supplementary Fig. 3B).

Representative fluorescence time traces exhibiting the fork regression activity of WRN was shown in Fig. 2h. A Cy3-Cy5 FRET jump occurred with a time delay after the delivery of ATP-Mg$^{2+}$ ($\tau_i$: time delay between ATP injection and FRET appearance). The high-FRET state was maintained for a while ($\tau_d$: time duration of high-FRET state before fluorescence signal disappearance) until the Cy3 and Cy5 signals simultaneously disappeared. We found that PCNA loading significantly reduced the fork regression activity of WRN (Fig. 2i), increased both $\tau_i$ (Fig. 2j and Supplementary Fig. 3C–E) and $\tau_d$ (Fig. 2k and Supplementary Fig. 3F–H). These data suggest that PCNA that remained on DNA at the replication forks could inhibit fork regression in ATAD5-depleted cells.

It has recently been reported that the interaction between RAD51 and the RAD51AP1-UAF1 complex is important for regulating DSB repair[21]. Since ATAD5 also interacts with UAF1, we investigated the effects of depletion of those proteins on replication forks under replication stress. Interestingly, we found that depletion of RAD51 or UAF1 reduced fork restart and the native BrdU signal under replication stress, but RAD51AP1 depletion did not (Fig. 2l, m and Supplementary Fig. 2E). In addition, UAF1 depletion reduced RAD51 recruitment to the stalled forks (Fig. 2n). These data suggest that while both RAD51AP1 and UAF1 are required for RAD51 recruitment to DSBs, RAD51AP1 is dispensable for RAD51 recruitment to stalled forks.

**ATAD5-RAD51 interaction increases under replication stress.** To find the molecular mechanisms of RAD51 recruitment to stalled forks, we examined the protein–protein interaction between ATAD5 and RAD51. We found that ATAD5 interacted with RAD51 and UAF1, but not with RAD51AP1 (Fig. 3a). The interaction of ATAD5 with RAD51 was increased by HU treatment, while the interaction with UAF1 was not affected. Similarly, the interaction of UAF1 with RAD51 was increased by HU treatment (Fig. 3b). Interestingly, an enhanced interaction of ATAD5 or UAF1 with RAD51 upon HU treatment was not observed in cells treated with the ATR inhibitor ETP-46464 (Fig. 3a, b). The ATR inhibitor also reduced RAD51 recruitment to stalled forks upon HU treatment as measured by iPOND,

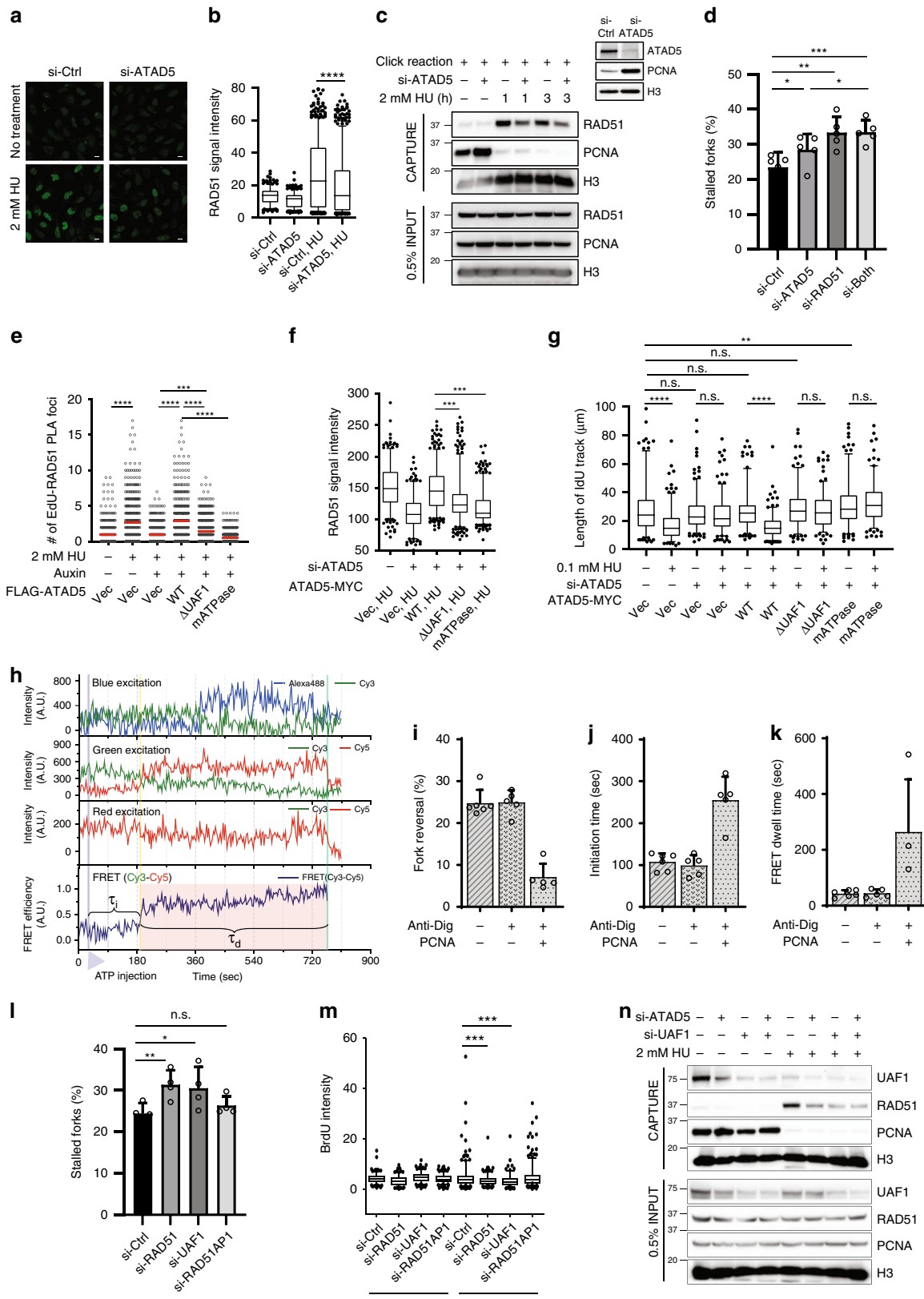

which was not affected by ATAD5 depletion (Fig. 3c). These results suggest that RAD51 recruitment to stalled forks is mediated by an interaction with ATAD5 in an ATR-dependent manner. We have examined several potential ATR phosphorylation sites in ATAD5 and RAD51, including those reported to be phosphorylated by Mec1, the ATR homolog in budding yeast[22], but these sites did not affect the interaction (Supplementary Table 1), suggesting that ATR regulates RAD51 recruitment through a complex mechanism.

We determined the RAD51-interacting regions in ATAD5. Both the N-terminal region (residues 600–700) and C-terminal region (residues 787–1844) of ATAD5 interacted with RAD51

**Fig. 2 ATAD5 promotes RAD51 recruitment to stalled replication forks, which depends on the PCNA unloading activity of ATAD5. a, b, d, l, m** U2OS cells were transfected with siRNA under the Noco-APH condition. **a, b** Cells were treated with 2 mM HU for 3 h before fixation for immunostaining. **a** Representative images of chromatin-bound RAD51. Scale bar: 20 μm. **b** The intensity of RAD51 staining was quantified. **c** After transfection, HEK293T cells were processed for iPOND immunoblotting. The right panel shows chromatin-bound proteins. **d** After transfection, cells were subjected for a DNA combing assay ($n = 5$). **e** U2OS cells expressing ATAD5$^{AID}$ transfected with a cDNA expression vector were subjected for a SIRF assay ($n = 3$ unless indicated). **f, g** U2OS-TetOn-ATAD5 cell lines were treated with doxycycline and transfected under the Noco-APH condition. **f** The staining intensity of chromatin-bound RAD51 was quantified. **g** Cells were analyzed using a DNA combing assay. The length of I-dU-labeled DNA track was measured. **h** Representative time traces of Alexa488 (blue), Cy3 (green), and Cy5 (red) at Alexa488 excitation (top panel), at Cy3 excitation (second panel), and at Cy5 excitation (third panel). The time trace of Cy3-Cy5 FRET at Cy3 excitation is presented at the bottom panel. Three events: ATP-Mg$^{2+}$ injection (blue), formation of a four-way junction (yellow), and dissociation of daughter strands (green) are indicated by color lines, respectively. Time delays between the events are defined as $\tau_i$ (initiation time) and $\tau_d$ (FRET dwell time, region with red shade). **i** Percentages of molecules that exhibit fork reversal activity ($n > 5$). **j** Average $\tau_i$ ($n > 5$). **k** Average $\tau_d$ ($n > 4$). **l** DNA combing assay ($n = 4$). **m** Native BrdU assay. **n** HEK293T cells transfected were processed for iPOND immunoblotting. **b, f, g, m** Boxes indicate median and interquartile ranges and whiskers indicate the 5th–95th percentile. Statistical analysis: Mann–Whitney U test. **d, e, l** Statistical analysis: t test; ****$p < 0.0001$, ***$p < 0.001$, **$p < 0.01$, *$p < 0.05$, n.s., not significant. **d, i–k, l** Error bars represent standard deviation of the mean.

(Fig. 3d). Interestingly, the former but not the latter region showed a HU-induced increase of interaction with RAD51. Further defined mapping of the N-terminal region of ATAD5 revealed that amino acid residues 642–645 of ATAD5 were important for interaction with RAD51 (Fig. 3e). An in vitro pull-down assay with purified proteins showed a direct interaction of residues 550–750 of ATAD5 with RAD51 (Fig. 3f). We also further mapped the C-terminal region of ATAD5 that mediates the interaction with RAD51. Residues 1630–1719 of ATAD5 were important for interaction with RAD51 (Fig. 3g). ATAD5 C-terminal fragments that were defective in interaction with RAD51 were also defective in interaction with RFC5, suggesting that the interaction between the C terminus of ATAD5 and RAD51 appears to be indirectly mediated by RFC2–5. Consistently, RFC4 depletion reduced the interaction between the C terminus of ATAD5 and RAD51 (Fig. 3h). Taken together, ATAD5 interacts with RAD51 both directly using an N-terminal region and indirectly using a C-terminal region.

We investigated whether protein–protein interaction between ATAD5 and RAD51 is important for RAD51 recruitment to stalled forks using the SIRF assay (Fig. 3i). In contrast to wild-type ATAD5, which restores HU-induced association RAD51 with nascent DNA, ATAD5 deletion mutant (Δ642–645), which can still unload PCNA (Fig. 3j), partially restored the HU-induced association of RAD51 with nascent DNA (Fig. 3i). This result suggests that the N-terminal region of ATAD5, which directly interacts with RAD51 is important for RAD51 recruitment to stalled replication forks under replication stress.

**ATAD5 promotes generation of ssDNA-associated breaks.** According to recent reports, stalled/regressed forks undergo various types of processing, leading to breakage, rearrangement, and recombination[1]. We examined the effect of ATAD5 depletion on DNA break formation under replication stress. HU treatment induced RPA2 S4/S8 phosphorylation, which is a marker of ssDNA-associated breaks[23,24]. The level of HU-induced RPA2 S4/S8 phosphorylation was reduced by ATAD5 depletion (Fig. 4a and Supplementary Fig. 4A). HU-induced nuclear RPA2 S4/S8 phosphorylation was also reduced upon ATAD5 depletion by siRNA or auxin treatment to cells expressing ATAD5$^{AID}$ (Fig. 4b–d). HU-induced RPA2 S4/S8 phosphorylation was decreased by treatment with a DNA-dependent protein kinase (DNA-PK) inhibitor NU7026 and the effect of the DNA-PK inhibitor was unchanged by ATAD5 depletion (Supplementary Fig. 4B). This observation indicates that DNA-PK is responsible for the change in HU-induced RPA2 S4/S8 phosphorylation that occurred with the depletion of ATAD5[24]. The expression of

siRNA-resistant wild-type ATAD5 in ATAD5-depleted cells restored the level of RPA2 S4/S8 phosphorylation, excluding indirect effects (Fig. 4e). Replication stress causes fork breakage at both ongoing replication forks and newly fired replication origins[25]. The reduction of HU-induced RPA2 S4/S8 phosphorylation by ATAD5 depletion was not affected by a CDC7 inhibitor that blocks new origin firing (Fig. 4f). This result suggests that the effect of ATAD5 on DNA breaks under replication stress is mainly exerted at ongoing replication forks. The iPOND assay results also showed reduced RPA2 S4/S8 phosphorylation at the stalled forks upon ATAD5 depletion (Fig. 4g). Collectively, ATAD5 facilitates ssDNA-associated breaks at ongoing replication forks under replication stress.

The ATPase domain and the UAF1 interaction domain of ATAD5 were both required for HU-induced RAD51 recruitment to stalled forks (Fig. 2e). We examined whether these domains are also important for formation/generation of ssDNA-associated breaks using U2OS-TetOn cell lines expressing wild-type or mutant ATAD5. Consistent with the results for RAD51 recruitment, the wild-type ATAD5 completely restored HU-induced RPA2 S4/S8 phosphorylation, but both mutant proteins failed to do so (Fig. 4h), suggesting that both PCNA unloading activity and UAF1 interaction of ATAD5 are important for the processing of stalled forks.

Since the effects of ATAD5 on stalled forks were mediated by RAD51 recruitment, we examined the effects of RAD51 activity on RPA2 S4/S8 phosphorylation using either siRNA-mediated RAD51 depletion or treatment with B02, which is a chemical that is known to inhibit RAD51 filament formation[26]. Since the phosphorylation levels of RPA2 S4/S8 are affected by the cell cycle profile, we first set up appropriate conditions for siRNA-mediated depletion or B02 treatment that did not affect the proportion of cells in S phase (Supplementary Fig. 4C, D). As expected, treatment with B02 blocked the HU-induced chromatin-bound RAD51 signal (Supplementary Fig. 4E, F). Using these conditions, we found that both RAD51 depletion and B02 treatment reduced HU-induced RPA2 S4/S8 phosphorylation (Fig. 4i). Consistent with our previous data (Figs. 2n and 3b), UAF1 depletion reduced HU-induced RPA2 S4/S8 phosphorylation, but RAD51AP1 depletion did not (Fig. 4i and Supplementary Fig. 4G). Since PCNA ubiquitination increases in both ATAD5- and UAF1-depleted cells[16], it is possible that ubiquitinated PCNA affects the stalled forks under replication stress. However, HU-induced RPA2 S4/S8 phosphorylation was unchanged in USP1-depleted cells despite high levels of PCNA ubiquitination (Fig. 4j), suggesting that the elevated abundance of ubiquitinated PCNA was not the reason for the reduction in ssDNA-associated breaks in ATAD5- and UAF1-depleted cells.

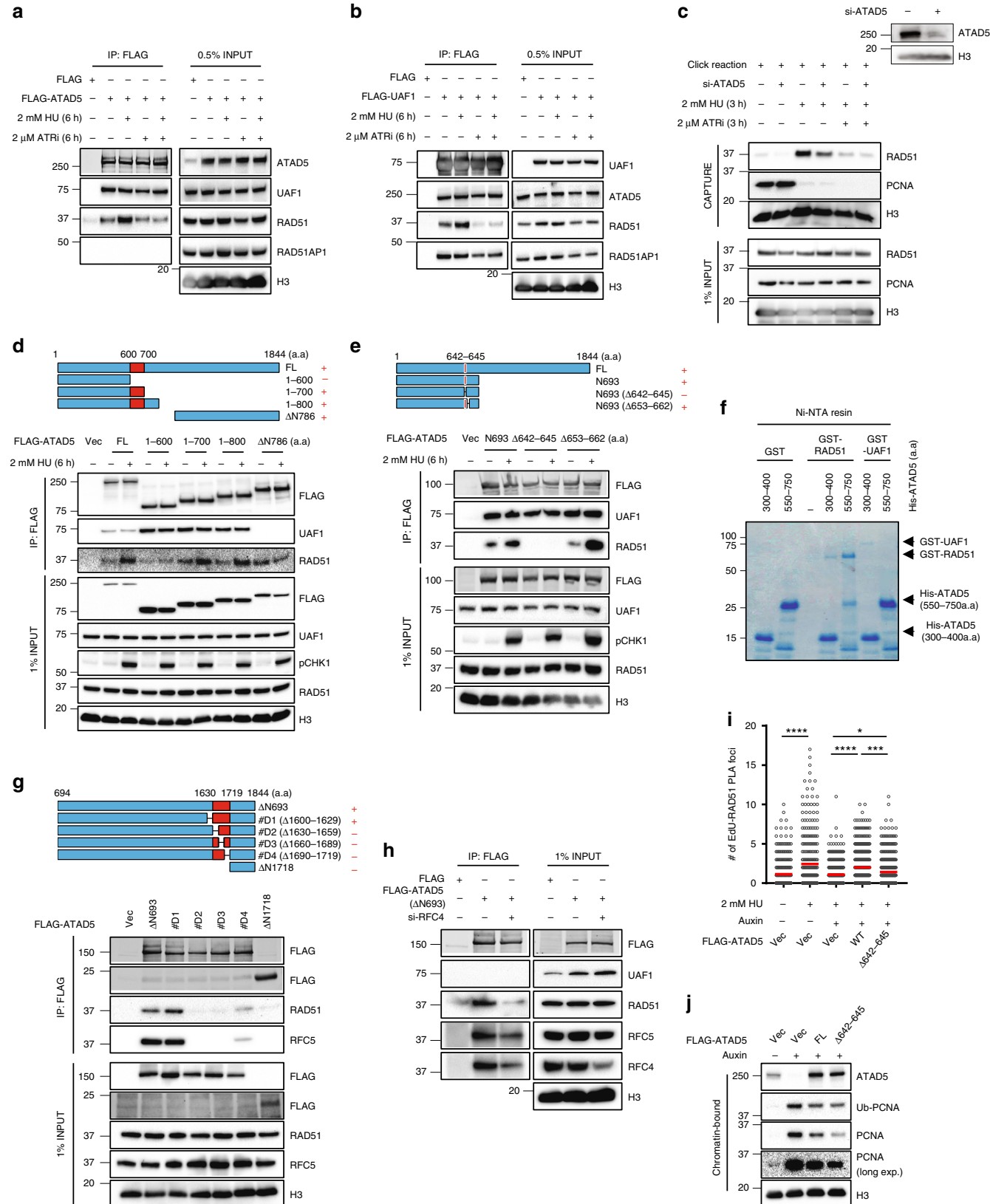

**ATAD5 promotes generation of MUS81-mediated DNA breaks**. We then assessed effects of ATAD5 depletion on physical DNA breaks under replication stress using the pulsed field gel electrophoresis. The intensity of break-containing DNA bands was increased by ~20% in wild-type cells following HU treatment, while no corresponding increase was observed

following ATAD5 depletion (Fig. 5a, b). We also measured DNA breaks at the single-cell level using neutral COMET (single-cell gel electrophoresis) assay, which detects DSBs. Here too, we observed a significant reduction in DSB formation in cells where ATAD5 was depleted by siRNA or AID as indicated by the COMET tail moment (Fig. 5c, d). However, in the alkaline

**Fig. 3 The interaction of ATAD5 with RAD51 is increased under replication stress. a–e, g, h** HEK293T cells were transfected with a cDNA expression vector or siRNA as indicated. After 48 h, cells were treated with 2 mM HU and whole-cell extracts were prepared for immunoprecipitation. **a, b** Cells were transfected with a cDNA expression vector expressing FLAG-tagged ATAD5 (FLAG-ATAD5) or FLAG-tagged UAF1 (FLAG-UAF1), or an empty vector (FLAG). After 48 h, cells were treated simultaneously with 2 mM HU or 2 μM ATR inhibitor (ATRi, ETP-46464) for 6 h. **c** HEK293T cells transfected with *ATAD5* siRNA for 48 h were labeled with 10 μM EdU for 20 min prior to the addition of 2 mM HU or 2 μM ATRi as indicated. Samples were processed for iPOND, and captured proteins were separated by SDS-PAGE and immunoblotted. The right panel shows chromatin-bound proteins extracted from a portion of cells in **c** separated by SDS-PAGE and immunoblotted. **d, e** Cells were transfected with an expression vector expressing full-length (FL) or deletion mutants of FLAG-ATAD5. A schematic diagram of the ATAD5 deletion mutants is shown on the top. Boxes in red represent RAD51-interacting regions. + indicates that the interaction exists. **f** Purified GST, GST-RAD51, or GST-UAF1 proteins were mixed with purified ATAD5 protein fragments and pulled down for Coomassie staining. **g** Cells were transfected with DNA vectors expressing deletion mutants of FLAG-ATAD5. A schematic diagram of the ATAD5 deletion mutants is shown on the top. Boxes in red represent RAD51-interacting regions. The symbol + indicates that the interaction exists. **h** Cells were transfected with a DNA vector expressing FLAG-tagged ATAD5 C-terminal fragment (ΔN693). After 6 h of transfection, cells were transfected with *RFC4* siRNA. **i** U2OS cells expressing ATAD5$^{AID}$ were transfected with a cDNA expression vector, treated with auxin, and fixed for a SIRF assay. Three independent experiments were performed and one representative result was displayed. Statistical analysis for **i**: *t* test; ****$p < 0.0001$, ***$p < 0.001$, *$p < 0.05$. **j** Chromatin-bound proteins extracted from a portion of cells in **i** were subjected for immunoblotting.

COMET assay, which detects both DSB- and single-strand breaks, we observed a similar COMET tail moment for the ATAD5-depleted and control cells (Supplementary Fig. 5A). Collectively, these results indicate that ATAD5 facilitates the occurrence of DSBs under replication stress.

We examined which activities of ATAD5 are important for the generation of DNA breaks using U2OS-TetOn cell lines. Consistent with the results for RPA2 S4/S8 phosphorylation (Fig. 4h), the wild-type ATAD5 restored HU-induced DNA breaks, but both UAF1 interaction-defective and ATPase-defective mutants of ATAD5 could not restore HU-induced DNA breaks (Fig. 5e). Thus, both the PCNA unloading activity and the UAF1 interaction of ATAD5 are important for the processing of stalled forks.

To find the responsible endonucleases for ATAD5-facilitated DNA breaks, we depleted several helicases and nucleases that had been reported to function at stalled/regressed forks and examined HU-induced RPA2 S4/S8 phosphorylation. We set up siRNA knockdown conditions that do not affect the proportion of cells in S phase to minimize the cell cycle effect (Supplementary Fig. 5B). The depletion of MUS81/EME2 reduced RPA2 S4/S8 phosphorylation (Fig. 5f and Supplementary Fig. 5C). SLX4, which is a scaffold protein for MUS81, also reduced RPA2 S4/S8 phosphorylation when depleted (Fig. 5f). Other nucleases such as CtIP, MRE11, and EXO1 did not affect RPA2 S4/S8 phosphorylation when depleted.

MUS81 endonuclease facilitates fork restart[10]. We found that HU-induced DNA breaks were reduced in MUS81-depleted cells (Fig. 5g). Depletion of both ATAD5 and MUS81 showed a similar level of reduction compared to a single depletion. In addition, we found that the abundance of MUS81 at the replication forks was reduced in ATAD5-depleted cells (Fig. 5h). The abundance of UAF1 was reduced at the replication forks upon HU treatment as previously reported[25]. Interestingly, the level of UAF1, but not that of RAD51AP1, was further reduced at the replication forks in ATAD5-depleted cells (Fig. 5h). We also found that SLX4 depletion, but not EME2 depletion, significantly reduced fork restart (Supplementary Fig. 5D). Taken together, ATAD5 promotes the generation of MUS81-mediated DNA breaks under replication stress, which is an alternative pathway to restart stalled/regressed forks.

**ATAD5 maintains genomic stability under replication stress.** Since ATAD5 is important for restarting stalled forks, the survival capacity of cells depleted for ATAD5 was analyzed using a colony survival assay. HU treatment reduced colony formation, and it was further reduced by ATAD5 depletion (Fig. 6a, b).

Next, we examined whether ATAD5 depletion affects genomic stability under replication stress by measuring chromosome breakage. HU treatment increased chromosome breakage, as previously reported, and it was further enhanced by ATAD5 depletion (Fig. 6c, d). The wild-type ATAD5 restored HU-induced chromosome breakage, but both ATPase-defective and UAF1 interaction-defective mutants of ATAD5 failed to restore it (Fig. 6e), which suggests that both activities are important for ATAD5 to maintain genomic stability. Next, we examined the effects of ATAD5 depletion on genomic stability in mice using the in vivo micronuclei assay. Reticulocytes in blood show an increased frequency of micronuclei when there is genomic instability[27]. The percentage of micronucleated reticulocytes increased after the intraperitoneal injection of 0.1 g/kg HU and peaked at 48 h (Fig. 6f). *Atad5*-haploinsufficient mice showed a significant increase in the percentage of micronucleated reticulocytes as compared to wild-type mice. Taken together, ATAD5 plays important roles in stabilizing and restarting stalled forks to maintain genomic integrity under replication stress.

**Discussion**

Replication stress alters the protein composition of replication forks, but many replisome proteins remain at stalled or even collapsed forks[28]. However, it is not well understood how those replisome proteins contribute to fork stability under replication stress. In this study, we provide evidence supporting a critical role for ATAD5-RLC, which is known to remove PCNA from chromatin, in functionally maintaining replication forks in response to replication stress (Fig. 6g). Our data strongly suggest that timely unloading of PCNA by ATAD5-RLC and the subsequent efficient recruitment of RAD51 to stalled forks, which is mediated by a HU-induced interaction with ATAD5, are important for RAD51 activity to regress stalled forks[5,7]. Interaction with UAF1 is also important for ATAD5 to regulate RAD51 recruitment to stalled forks, independently of USP1-mediated PCNA deubiquitination. Unlike HR-mediated DSB repair, which requires a trimeric complex containing UAF1, RAD51AP1, and RAD51[21], RAD51AP1 is dispensable for RAD51 recruitment to stalled forks. This observation suggests that different protein requirements apply depending on the type of genotoxicity or the stage of the cell cycle. It is unclear whether ATAD5 plays a role in RAD51 loading. Alternatively, RAD51 recruitment and loading can be separate processes and can be assisted differently.

We have provided evidence suggesting that PCNA unloading by ATAD5 is required for the activity of RAD51. However, it remains unclear how PCNA unloading can positively affect RAD51 recruitment or RAD51 filament formation under replication stress. A recent report showed a correlation between

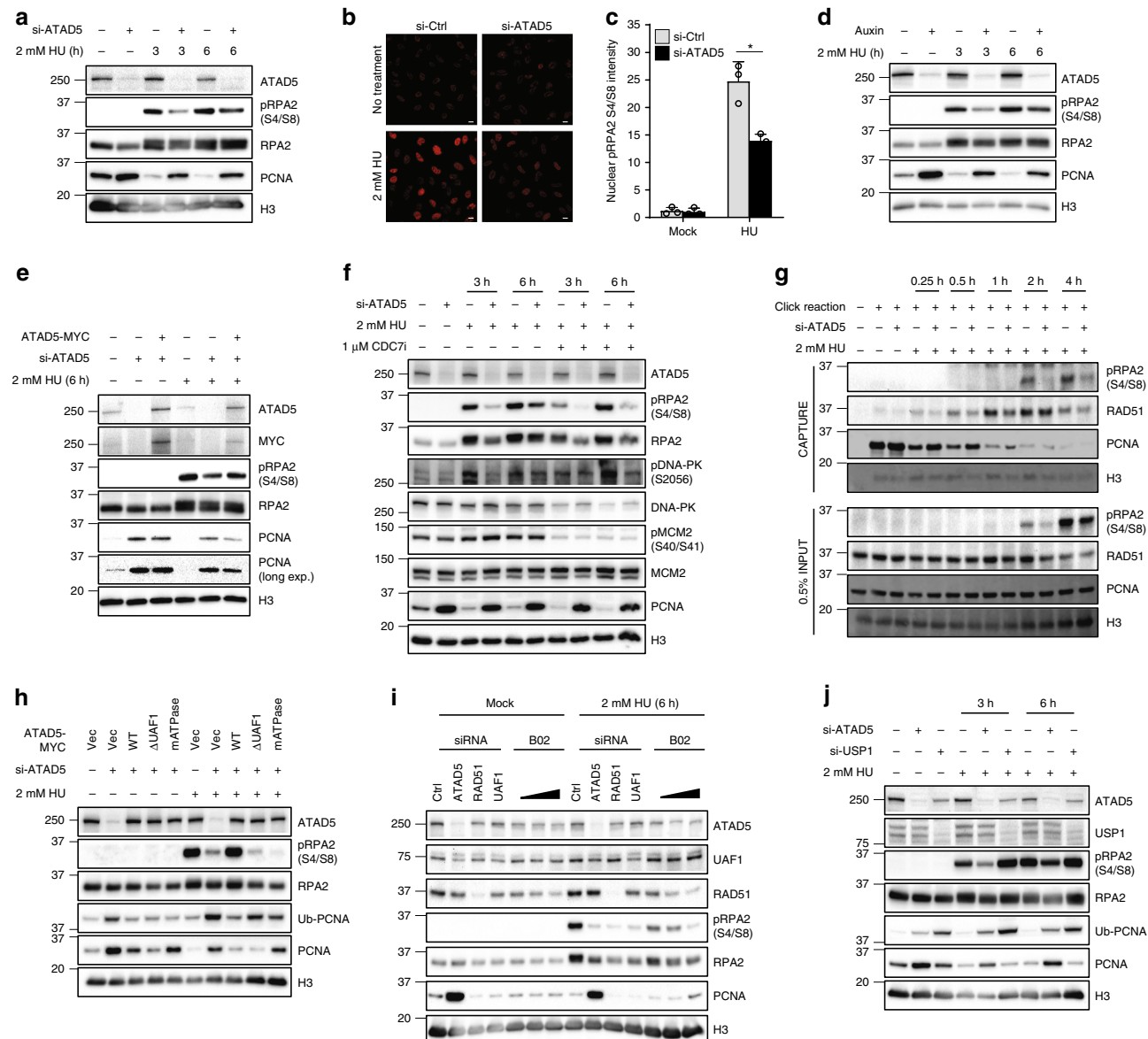

**Fig. 4 ATAD5 promotes generation of single-stranded DNA-associated breaks in response to replication stress. a** U2OS cells transfected with *ATAD5* siRNA under the Noco-APH condition were treated with 2 mM HU for 3 or 6 h. Then, chromatin-bound proteins were fractionated and subjected for immunoblotting. **b, c** U2OS cells transfected with *ATAD5* siRNA under the Noco-APH condition were treated with 2 mM HU for 3 h before fixation. The fixed cells were stained with an anti-pRPA2 S4/S8 antibody. **b** Representative images of chromatin-bound pRPA2 S4/S8. Scale bar: 20 μm. **c** The intensity of chromatin-bound pRPA2 S4/S8 staining was quantified from ~20,000 cells. Error bars represent standard deviation of the mean ($n = 3$). Statistical analysis: $t$ test; *$p < 0.05$. **d** U2OS cells expressing ATAD5[AID] were pre-treated with auxin and treated with 2 mM HU for 3 or 6 h. Chromatin-bound proteins were separated by SDS-PAGE and subjected for immunoblotting with indicated antibodies. **e** U2OS cells transfected with a combination of *ATAD5* siRNA and a DNA vector expressing ATAD5-myc under the Noco-APH condition were treated with 2 mM HU for 6 h. Then, chromatin-bound proteins were fractionated and subjected for immunoblotting. **f** U2OS cells transfected with *ATAD5* siRNA under the Noco-APH condition were treated with 2 mM HU or 1 μM CDC7 inhibitor (CDC7i, PHA-76941) for the indicated times. Then, chromatin-bound proteins were fractionated and subjected for immunoblotting. **g** HEK293T cells transfected with *ATAD5* siRNA for 48 h were labeled with 10 μM EdU for 20 min prior to treatment with 2 mM HU as indicated. Samples were processed for iPOND, and captured proteins were separated by SDS-PAGE and immunoblotted. **h** U2OS-TetOn-ATAD5 cells treated with doxycyclinfor 24 h before entering the Noco-APH condition were treated with 2 mM HU for 6 h. Chromatin-bound proteins were fractionated and subjected for immunoblotting. **i, j** U2OS cells transfected with siRNAs or treated with a RAD51 inhibitor (B02, 10, 20, 40 μM) at the time of release from aphidicolin under the Noco-APH condition were treated with 2 mM HU as indicated. Chromatin-bound proteins were fractionated and subjected for immunoblotting.

PCNA removal and RPA protein loading at DNA damage sites[29]. Similarly, it is possible that when ATAD5 is depleted, PCNA accumulated on the lagging strand at the replication forks might prevent the RAD51 filament from efficiently forming on the lagging strand. Otherwise, strand invasion by the RAD51 filament formed on the leading strand into homologous DNA on the lagging strand could be inhibited by remaining PCNA on the lagging strand. Another possibility is that PCNA itself might prevent the reannealing of single-stranded DNA at the beginning of fork regression. The reannealing of nascent DNA to generate a chicken foot-shaped four-way structure could also be affected by the remaining PCNA. The results of single-molecule experiments

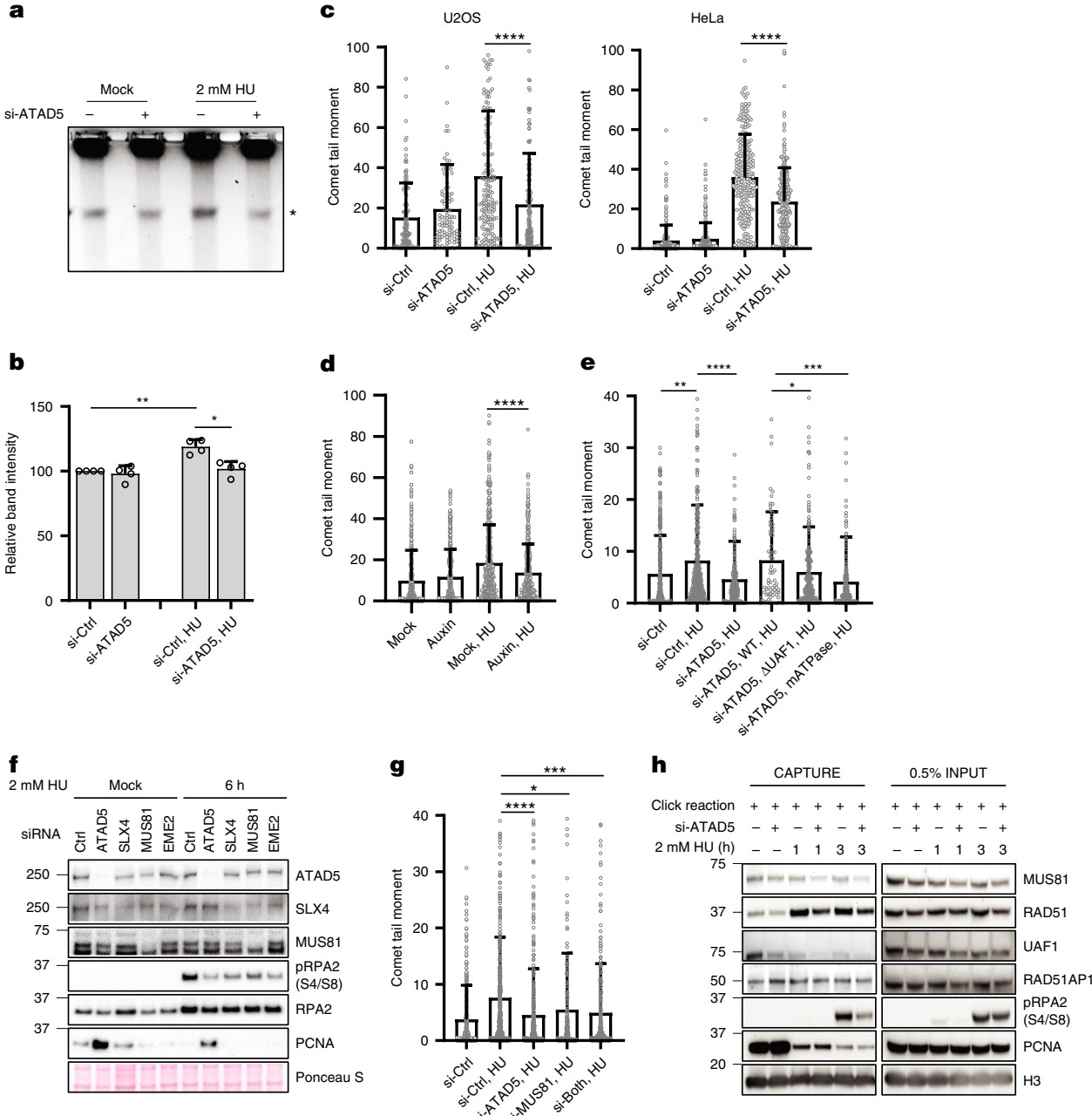

**Fig. 5 ATAD5 promotes generation of MUS81-mediated single-stranded DNA-associated breaks in response to replication stress. a**, **b** U2OS cells transfected with *ATAD5* siRNA under the Noco-APH condition were treated with 2 mM HU for 6 h before being collected for analysis by pulsed field gel electrophoresis. **a** Representative data from three independent experiments. Asterisk (*) indicates a DNA break. **b** DNA breaks were quantified and displayed (*N* = 4). **c** U2OS or HeLa cells transfected with *ATAD5* siRNA under the Noco-APH condition were treated with 2 mM HU for 6 h before being collected for a neutral COMET assay. The tail moment was calculated from ~200 cells and plotted. **d** U2OS cells expressing ATAD5^AID were pre-treated with auxin and treated with 2 mM HU for another 6 h before being collected for a neutral COMET assay. Two independent experiments were performed, and one representative result is displayed. **e** U2OS-TetOn-ATAD5 cells treated with doxycycline for 24 h before entering the Noco-APH condition were treated with 2 mM HU for 6 h before collection for the neutral COMET assay. **f** U2OS cells transfected with siRNAs under the Noco-APH condition were treated with 2 mM HU for 6 h. Chromatin-bound proteins were fractionated and subjected to immunoblotting. **g** U2OS cells transfected with a combination of *ATAD5* and *MUS81* siRNAs under the Noco-APH condition were treated with 2 mM HU for 6 h before collection for the neutral COMET assay. The tail moment was calculated from ~300 cells and plotted. **h** HEK293T cells transfected with *ATAD5* siRNA for 48 h were labeled with 10 μM EdU for 20 min, washed, and treated with 2 mM HU for the indicated times. Samples were processed for iPOND, and captured proteins were separated by SDS-PAGE and immunoblotted. **b**–**e**, **g** Error bars represent standard deviation of the mean. Statistical analysis: two-tailed Student's *t* test; *$p < 0.05$, **$p < 0.005$, ***$p < 0.001$, and ****$p < 0.0001$.

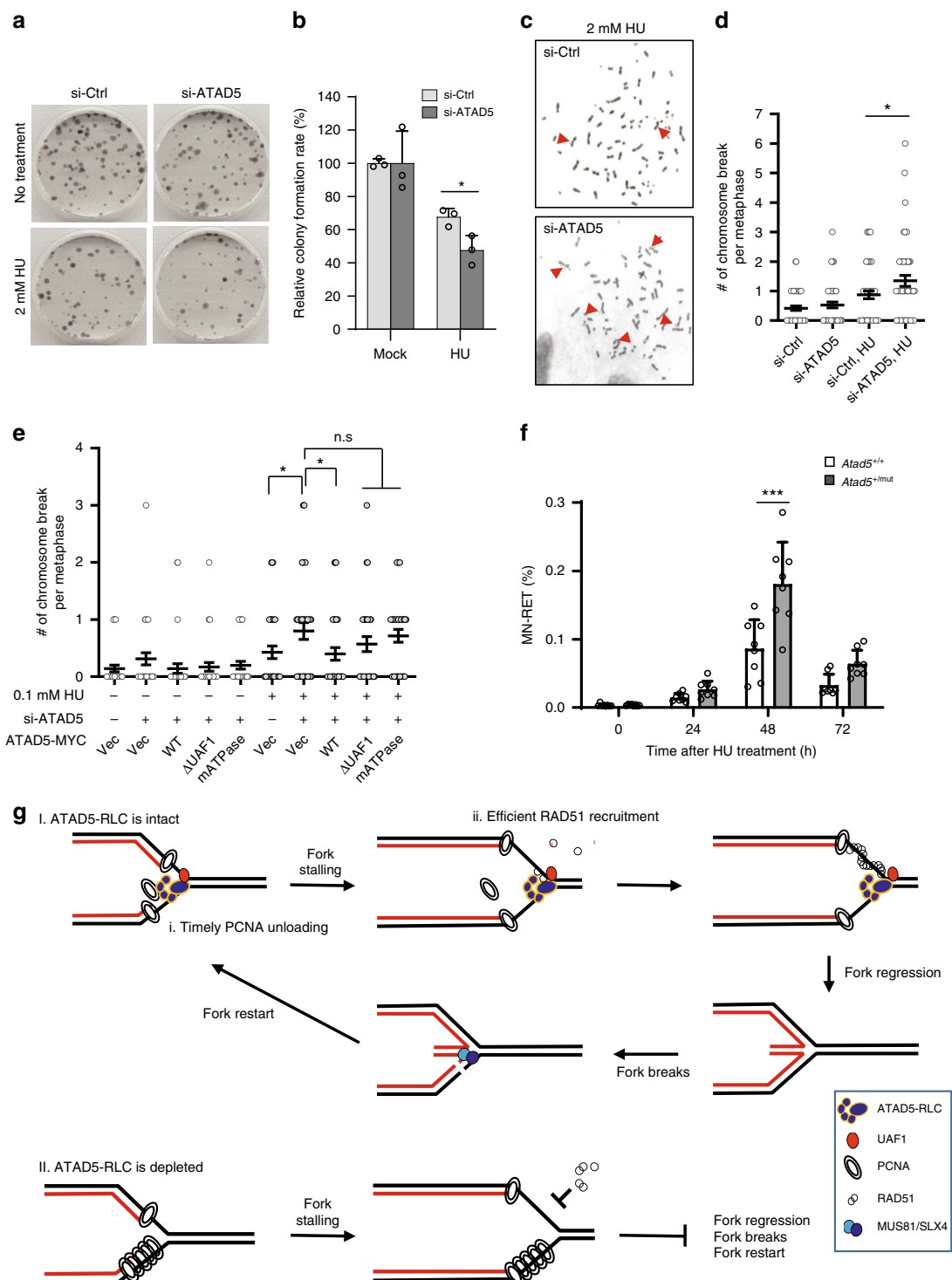

**Fig. 6 ATAD5 depletion increases cell death and genomic instability in cells and mice. a**, **b** U2OS cells transfected with *ATAD5* siRNA under the Noco-APH condition were treated with 2 mM HU for 6 h, plated in the absence of drug treatments, incubated for 2 weeks, and examined to detect surviving colonies by methylene blue staining. **a** Representative images of colony staining. **b** The viability of the surviving colonies was quantified. Error bars represent standard deviation of the mean ($n = 3$). Statistical analysis: two-tailed Student's *t* test; *$p < 0.05$. **c**, **d** U2OS cells transfected with *ATAD5* siRNA under the Noco-APH condition were treated with HU for 6 h and subjected to metaphase spreading. **c** Representative images of DAPI stained chromosomes are shown. Arrows point to chromosomal breaks. **d** The number of chromosomal breaks per cell was plotted. Statistical analysis: two-tailed Student's *t* test; *$p < 0.05$. **e** U2OS-TetOn-ATAD5 cell lines expressing wild-type or two ATAD5 mutants were treated with doxycycline and transfected with *ATAD5* siRNA under the Noco-APH condition. Cells were then treated with HU for 6 h and subjected to metaphase spreading. Statistical analysis: two-tailed Student's *t* test; *$p < 0.05$. n.s., not significant. **f** *Atad5* wild-type or mutant mice were analyzed by micronucleus analysis. Error bars represent standard deviation of the mean ($n = 8$). Statistical analysis: two-way analysis of variance, ***$p < 0.001$. **g** The model described in the text.

support that possibility (Fig. 2h–k). However, to more clearly understand the relationship between PCNA regulation and RAD51-mediated fork regression, several issues still need to be addressed, including how RAD51 recruitment and loading are coordinated and where PCNA accumulates among ssDNA or dsDNA regions.

PCNA is mono- or poly-ubiquitinated on lysine 164 when the progression of the replication fork is blocked[30]. Recently, it was reported that the recruitment of DNA translocase ZRANB3 to stalled forks by recognizing poly-ubiquitin chains, presumably on PCNA, under various DNA damage conditions stimulates fork regression[31]. ATAD5 depletion increases the abundance of mono-ubiquitinated PCNA on the chromatin owing to a reduction in the activity of USP1 against PCNA[16]. Poly-ubiquitinated PCNA is also likely to increase in ATAD5-depleted cells. Conceptually, fork regression through this mechanism appears to be different from our observations. First, our experimental conditions for replication stress induced mainly PCNA mono-ubiquitination since poly-ubiquitinated PCNA was not detected under our conditions even in USP1-depleted cells (Fig. 4j). In addition, based on our iPOND-mass spectrometry data, the recruitment of ZRANB3 to the stalled forks was not detected under our HU treatment condition. The recombinase RAD51 and the translocase ZRANB3 can cooperate to drive fork regression, but depending on the type of genotoxicity, one of them may be dominant. Indeed, ZRANB3-depleted cells displayed different sensitivities following genotoxic treatments[32].

ATR kinase is a master regulator that processes replication stress[33]. ATR senses stalled replication forks and coordinates multiple replication stress responses to arrest the cell cycle, block new origin firing, and stabilize/restart stalled forks. Focusing specifically on fork stability, ATR regulates a translocase SMARCAL1 by phosphorylation to balance the amount of fork regression[34]. We found that the HU-induced interaction between ATAD5 and RAD51 was regulated by ATR (Fig. 3a), which suggests that the phosphorylation of either protein by ATR could enhance the interaction between them. The proteins Elg1 and Rad51 were previously reported to be phosphorylated by Mec1, which is the ATR homolog in budding yeast[22,35]. In the present work, we found that the HU-induced interaction between ATAD5 and RAD51 was unaffected by mutations of amino acids corresponding to those phosphorylation sites. We also checked other SQ/TQ sites in ATAD5 around the RAD51 interaction region, but those sites did not seem to affect the interaction between ATAD5 and RAD51. The mechanisms that regulate the interaction between ATAD5 and RAD51 seem to be complex and may involve multiple phosphorylation events on multiple proteins.

Regressed forks can be restored to replicate again through several mechanisms such as RECQ1-mediated branch migration and DNA2-mediated controlled resection of the regressed arm[1]. MUS81/SLX4-mediated fork cleavage has been suggested to be required for the DNA synthesis of common fragile sites during mitosis[36–38]. Accumulating data support the positive effects of the structure-specific endonuclease MUS81 for fork restart during DNA replication[10,39,40], although a prior report claimed that MUS81-mediated DNA DSB led to cell death following replication stress[41]. Variations in the dose and duration of treatment could produce different outcomes. We found that ATAD5 depletion reduced HU-induced RPA2 S4/S8 phosphorylation and DNA breaks (Figs. 4 and 5). The depletion of MUS81, EME2, or SLX4 also reduced HU-induced RPA2 S4/S8 phosphorylation (Fig. 5f), suggesting that our replication stress condition generates stalled replication intermediates that need to be resolved by MUS81/SLX4-mediated endonucleolytic cleavage.

ATAD5 is considered to function as a tumor suppressor based on the tumor incidence in *Atad5* heterozygote mutant mice and frequent mutations of the *ATAD5* gene in patients with various types of cancer[13,14]. Considering the molecular mechanism of ATAD5, defects in two PCNA-regulating functions, that is, PCNA deubiquitination and PCNA unloading, could be sources of tumorigenesis in ATAD5-compromised cells. The former might contribute to the reduction of mutagenesis by restricting error-prone translesion synthesis activity[16]. Here, we have provided another molecular mechanism that might explain the tumor-suppressive function of ATAD5. It should be emphasized that multiple regulatory methods, including protein–protein interactions, are used by ATAD5 to maintain genomic stability.

## Methods

**Cell lines.** Human embryonic kidney (HEK) 293T (ATCC® CRL-3216™), HeLa (ATCC® CCL-2™), and U2OS (ATCC® HTB-96™) cells were cultured in Dulbecco's modified Eagle's medium containing 10% fetal bovine serum (GE Healthcare, Little Chalfont, UK), 100 U/mL penicillin G (Life Technologies, Carlsbad, CA), and 100 µg/mL streptomycin (Life Technologies). U2OS cells expressing osTIR1-9Myc were generated by retroviral infection of pBabe Blast osTIR1-9Myc from Andrew Holland (Addgene, plasmid # 80073). To generate ATAD5^{AID} cell line, U2OS-osTIR1-9Myc cells were co-transfected with CRISPR/Cas9, single guide RNA (see Supplementary Table 2 for the sequence information) and donor plasmids by nucleofection (Lonza). Forty-eight hours after transfection, high green fluorescent protein (GFP)-expressing cells were sorted into 96-well plates using a FACSAria Fusion (BD Biosciences). Genomic DNA PCR was used to identify positive clones. Indole-3-acetic acid (500 µM), a natural auxin, was added to the culture medium to induce degradation of AID-tagged ATAD5. To conditionally overexpress ATAD5, a Lenti-X™ TetOn® 3G-inducible expression system was used following the manufacturer's protocol (Clontech Laboratories, Mountain View, CA). Briefly, the wild-type ATAD5 complementary DNA (cDNA) and mutant ATAD5 cDNAs either with a defect in UAF1 interaction or the ATPase activity (an E1173K mutation) were cloned into the pLVX-TRE3G-ZsGreen1 vector and viral particles were produced. The viral particles were infected into U2OS cells expressing Tet3G and selected by puromycin. To induce protein expression, doxycycline (final concentration 100 ng/mL; Sigma-Aldrich, St. Louis, MO) was administered.

**Plasmids.** For the construction of a CRISPR/Cas9 plasmid targeting the endogenous *ATAD5* locus, we used pSpCas9n(BB)-2A-GFP from Feng Zhang (Addgene, plasmid #48140) according to the previous protocol[42]. Briefly, a pair of complementary oligomers targeting the region near the ATG start codon of the endogenous *ATAD5* locus were annealed and ligated to the *Bbs*I-digested pSpCas9n(BB)-2A-GFP vector. To generate a donor plasmid for tagging endogenous ATAD5 with mini auxin-inducible degron (mAID), genomic DNA was amplified with homology arms to the target locus (about 250 bp each) using Phusion DNA polymerase (New England Biolabs) and cloned into a pGEM T-easy vector (Promega). The mAID cassette was then cloned between the ATG codon and the second codon of *ATAD5* gene of the donor plasmid.

**Chemicals.** The following drugs were used in this study: HU, aphidicolin, nocodazole (Sigma-Aldrich), ATR inhibitor (ETP-46464; Selleckchem, Houston, TX), DNA-PK inhibitor (Nu7026; Selleckchem), CDC7 inhibitor (PHA-767491; Selleckchem), RAD51 inhibitor (B02; Merck Millipore, Burlington, MA), and MRE11 inhibitor (mirin; Sigma-Aldrich).

**Antibodies.** The following antibodies were used: anti-α-tubulin (Sigma-Aldrich, T9026, 1:10,000); anti-PCNA (Santa Cruz Biotechnology, sc-56, 1:2000); anti-UAF1 (Santa Cruz Biotechnology, sc-514473, 1:500); anti-RFC4 (Santa Cruz Biotechnology, sc-20996,1:1000); anti-RFC5 (Santa Cruz Biotechnology, sc-20997, 1:1500); anti-LAMIN B1 (Santa Cruz Biotechnology, sc-20682, 1:2500); anti-CHK1 (Santa Cruz Biotechnology, sc8408, 1:500); anti-pCHK1(S317) (Bethyl, A304-673A, 1:5000); anti-CHK2 (GeneTex, GTX70295, 1:1000); pCHK2(T68) (Cell Signaling Technology, #2197, 1:1000); anti-RPA2 (Bethyl, A300-244A, 1:3000); anti-pRPA2(S4/S8) (Bethyl, A300-245A, 1:3000); anti-pRPA2(S33) (Bethyl, A300-246A, 1:3000); anti-RAD51 (Cell Signaling Technology, #8875, 1:1000); anti-Ub-PCNA (Cell Signaling Technology, #13439, 1:1000); anti-MYC (Merck Millipore, 05-724.1:100); anti-FLAG (Sigma-Aldrich, F3165, 1:1000); anti-MCM2 (Abcam, ab4464, 1:2000); anti-pMCM2(S40/S41) (Bethyl, A300-788A, 1:2000); anti-MUS81 (Abcam, ab14387, 1:1000); anti-SLX4 (Abcam, ab169114, 1:1000); anti-DNA-PK (Thermo Scientific, MS-423-P1, 1:1000); anti-pDNA-PK(S2056) (Abcam, ab18192, 1:1000); anti-histone H3 (Merck Millipore, 07-690, 1:10,000); anti-γH2AX (Merck Millipore, 05-636, 1:2000); anti-MRE11 (Novus Biologicals, NB100-142, 1:1000); anti-ATM (Santa Cruz Biotechnology, sc-23921, 1:1000); anti-pATM(S1981) (R&D Systems, AF1655, 1:1000); anti-RAD51AP1 (GeneTex, GTX115455, 1:1500). The

anti-human ATAD5 antibody was raised in rabbits using the N-terminal fragment (residues 1–297 amino acids)[17].

**Transfections and RNA interference**. Transfections of plasmid DNA and siRNAs, either synthetic duplexes or SMART pool (20 nM), were performed using X-tremeGENE™ HP (Roche, Basel, Switzerland) and RNAiMAX (Thermo Fisher Scientific), respectively, according to the manufacturer's instructions with slight modifications. The transfection reagent was removed 5 h after transfection and fresh medium was added. See Supplementary Table 2 for the siRNA sequence information.

**Immunoprecipitation and immunoblot analysis**. A Triton X-100™-soluble fraction (soluble fraction) and a Triton X-100™-insoluble fraction (chromatin-bound fraction) were isolated and subjected for immunoblot analysis according to the methods described previously[17] with slight modifications. In brief, the soluble fraction was isolated by incubating cells in buffer A (100 mM NaCl, 300 mM sucrose, 3 mM MgCl₂, 10 mM PIPES, pH 6.8, 1 mM EGTA, 0.2% Triton X-100™, phosphatase inhibitors, and protease inhibitors [Roche]) for 5 min on ice, followed by centrifugation. Then, the chromatin-bound fraction was isolated by resuspending the pellet in RIPA buffer (50 mM Tris-HCl, pH 8.0, 150 mM NaCl, 5 mM EDTA, 1% Triton X-100™, 0.1% sodium dodecyl sulfate, 0.5% sodium deoxycholate, 0.1 M PMSF, phosphatase inhibitors, and protease inhibitors) with Benzonase® nuclease for 40 min on ice, followed by sonication and centrifugation. For immunoprecipitation, cells were lysed on ice in buffer X (100 mM Tris-HCl, 250 mM NaCl, 1 mM EDTA, 1% NP-40, 0.1 M phenylmethylsulfonyl fluoride, phosphatase inhibitors, and protease inhibitors) with Benzonase® nuclease, followed by sonication and centrifugation. Proteins were separated by sodium dodecyl sulfate-polyacrylamide gel electrophoresis (SDS-PAGE) and transferred to a nitrocellulose membrane. Blocking of the membranes and blotting with primary antibodies were performed in Tris-buffered saline containing 0.1% Tween® 20 supplemented with 5% skim milk powder. Proteins were visualized using secondary horseradish peroxidase-conjugated antibodies (Enzo Life Sciences, New York, NY) and enhanced chemiluminescence reagent (Thermo Fisher Scientific). Signals were detected using an automated imaging system (ChemiDoc™; Bio-Rad Laboratories, Hercules, CA).

**His-tag protein pull-down assay**. For Ni-NTA pull-down assays, Ni-chelated beads (Sigma-Aldrich) were equilibrated with binding buffer A (50 mM Tris-HCl, pH 7.5–8.0, 50 mM NaCl, 10% glycerol, 0.1% Triton X-100™, and 1 mM dithiothreitol) three times. Combined equilibrated beads and purified proteins (His-tagged ATAD5 proteins and purified GST-RAD51 or GST-UAF1) in a buffer that contains 10% glycerol and 0.1% Triton X-100™ then incubated for 1 h at 4 °C with rotation. The beads were then washed five times with 1 mL of binding buffer A supplemented with 20 mM imidazole. Bound proteins were collected from the nickel beads by adding elution buffer (buffer A and 250 mM imidazole) for 20 min at 4 °C. The eluted proteins were mixed with 2× Laemmli sample buffer, followed by incubation at 95 °C for 5 min. Proteins were analyzed by SDS-PAGE followed by Coomassie Brilliant Blue staining.

**DNA combing analysis**. Exponentially growing cells were labeled with 100 μM Cl-dU for 20 min, washed, treated with 2 mM HU for 2 h for fork restart analysis, and labeled with 250 μM I-dU for 30 min. To analyze HU-induced deceleration of replication fork progression, cells were pre-labeled with Cl-dU for 30 min, washed and labeled with I-dU, and treated with 0.1 mM HU for 30 min. Cells were harvested by trypsinization and embedded in a low-melting agarose plug at a density of $2.5 \times 10^5$ cells/plug. The plugs were lysed in lysis buffer (100 mM EDTA, 1% N-lauroyl-sarcosine, 10 mM Tris-Cl, pH 8.0, 1 mg/mL proteinase K) and melted at 68 °C for 20 min in the presence of 0.5 M 2-(N-morpholino)-ethanesulfonic acid (pH 5.5) with β-agarase. Samples were cooled down to about 42 °C. After β-agarase digestion, DNA was combed on phosphate-buffered saline (PBS) containing 0.1% Tween® 20, Alexa Fluor®-conjugated secondary antibodies (Thermo Fisher Scientific) were added and incubated for 45 min at room temperature (RT). Slides were mounted using ProLong® Gold antifade reagent (Vector Laboratories, Burlingame, CA). Confocal images were acquired with an LSM880 confocal microscope (Carl Zeiss, Oberkochen, Germany). Image acquisition and analysis were performed with ImageJ (National Institutes of Health, Bethesda, MD).

**iPOND assay**. The iPOND assay was performed as previously described[43] with slight modifications. HEK293T cells were transfected with siRNA for 48 h and incubated with 20 μM EdU for 20 min. Cells were then treated with 2 mM HU for 1 or 3 h and subsequently fixed using 1% formaldehyde for 20 min at RT. The crosslinking reaction was quenched using 0.125 M glycine and the cells were washed three times with PBS. Cells were incubated with 0.25% Triton X-100™ in PBS for 30 min at RT and were pelleted. Permeabilization was stopped with 0.5% bovine serum albumin in PBS. Cells were pelleted again and washed with PBS. After centrifugation, cells were resuspended with a click reaction cocktail and incubated for 1 h at RT on a rotator. After centrifugation, the click reaction was stopped by resuspending cells in PBS containing 0.5% serum bovine albumin. Cells were then pelleted and washed with PBS twice. Cells were resuspended in lysis

buffer and sonicated. Lysates were cleared and then incubated with streptavidin-agarose beads overnight at 4 °C in the dark. The beads were washed once with lysis buffer, once with 1 M NaCl, and then twice with lysis buffer. To elute proteins bound to nascent DNA, the 2× sodium dodecyl sulfate Laemmli sample buffer was added to packed beads (1:1; v/v). Samples were incubated at 95 °C for 30 min, followed by immunoblotting.

**SIRF assay**. The quantitative in situ analysis of protein interactions at DNA replication forks (SIRF) assay was performed as previously described[18] with a slight modification. Cells were plated on LabTek™ chamber slides (Thermo Fisher Scientific) and incubated with 100 μM EdU for 10 min. For replication stress conditions, EdU was removed and slides were washed two times with PBS before incubation in pre-warmed media with 2 mM HU for 3 h. Cells were then fixed with 2% paraformaldehyde in PBS (pH 7.4) for 20 min at RT and permeabilized with 0.25% Triton X-100 in PBS for 30 min at RT. Cells were washed with PBS twice for 5 min each. The click reaction cocktail (2 mM CuSO₄ (copper sulfate), 20 μM biotin azide and 10 mM sodium ascorbate in PBS) was added to each chamber and cells were incubated at RT for 30 min. Cells were then blocked in 10% fetal bovine serum in PBS for 1 h at RT. Cells were incubated with primary antibodies overnight at 4 °C (1:250 mouse anti-biotin antibody (sc-57636, Santa cruz) with 1:250 rabbit anti-RAD51 antibody (#8875, Cell Signaling)). Cells were washed twice with PBS and incubated with pre-mixed Duolink PLA plus and minus probes for 1 h at 37 °C. The subsequent steps in proximal ligation assay were carried out using the Duolink® PLA Fluorescence Kit (Sigma) according to the manufacturer's instructions. Slides were stained with DAPI (4′,6-diamidino-2-phenylindole) and imaged by a Zeiss LSM880 confocal microscope.

**Native BrdU assay**. The native BrdU assay was performed as previously described[34]. Briefly, the cells were treated with 10 μM BrdU for 10 min and washed in fresh media before treatment with 2 mM HU for 4 h. Cells were pre-extracted with CSK buffer (10 mM PIPES, 100 mM NaCl, 300 mM sucrose, 3 mM MgCl₂, 1 mM EGTA, and 0.5% Triton X-100™) for 10 min, fixed with 4% formaldehyde, and immunostained with an anti-BrdU antibody (BD Biosciences, Franklin Lakes, NJ) without a DNA denaturation step. Images were acquired using an LSM880 confocal microscope (Carl Zeiss). The mean BrdU intensity per nucleus was scored for each sample using the ZEN Blue software (Carl Zeiss). Statistical analysis was performed using the Prism 6 software (GraphPad, La Jolla, CA).

**Pulsed field gel electrophoresis**. For DNA break analysis, $10^6$ cells were mixed into melted agarose inserts. The agarose inserts were incubated in proteinase K buffer (0.5 M EDTA, 1% N-laurylsarcosyl, and 1 mg/mL proteinase K) at 50 °C for 48 h, and thereafter, washed four times in Tris-EDTA buffer prior to loading onto a 1% agarose gel (chromosomal grade; Bio-Rad Laboratories) and separated using a CHEF DR III pulsed field gel electrophoresis apparatus for 24 h (Bio-Rad Laboratories; 120° field angle; 240 s switch time; 4 V/cm at 14 °C). The gel was subsequently stained with ethidium bromide and analyzed with ImageJ.

**COMET assay**. The COMET assay was performed using a CometAssay® Kit (Trevigen, Gaithersburg, MD) according to the manufacturer's instructions. In brief, each cell suspension was mixed with COMET LMAgarose at 37 °C and the mixture was spread on a COMET slide (Trevigen). After the solidification of the agarose, the slide was immersed in a lysis solution (Trevigen) for 1 h at 4 °C. For both neutral and alkaline COMET assays, images were acquired with a fluorescence microscope (BX53; Olympus, Tokyo, Japan) and the tail moment was calculated using the CometScore software version 2.0.

**Preparation of DNA substrates**. DNA strands were purchased from Integrated DNA Technologies (IDT, Coralville, IA) (see Supplementary Table 3 for the sequence information). For dye labeling, 0.5 mM DNA modified with an amino C6 dT was incubated with 10 mM amine-reactive fluorophore in a reaction buffer (100 mM Na₂BO₇ [pH 8.5]) for 6 h. The excess dye was removed using ethanol precipitation. Model replication forks were sequentially annealed as follows: first, the leading and lagging arms were annealed separately by cooling the mixture of the parent and daughter strands from 90 °C to 4 °C at a rate of −1 °C/min; next, the leading and lagging arms were mixed and cooled from 50 °C to 4 °C at a rate of −1 °C/min. The annealing reaction was performed in a 10 mM Tris-HCl (pH 8.0) buffer containing 50 mM NaCl.

**Single-molecule fork regression assay**. Quartz slides and glass coverslips were cleaned with piranha solution (mixture of sulfuric acid and hydrogen peroxide) and then coated with a 40:1 mixture of polyethylene glycol (PEG) and biotin-PEG. A simple microfluidic sample chamber (volume: ~20 μl) was made by assembling the PEG-coated quartz slide and a glass coverslip using double-sided tape. Plastic tubing and a syringe pump (Fusion 100, Chemyx) were used for automatic buffer exchange during measurements. A flow rate of 4000 μl/min was used so that buffer exchange time was less than the time resolution of the experiment (0.4 sec vs. 2.0 s,

respectively). The DNA samples were immobilized on the PEG-passivated surface via biotin–streptavidin interactions and then incubated with 100 nM anti-digoxigenin (Roche) for 20 min, followed by incubation with 12 nM Alexa488-labeled PCNA in loading buffer (Saturated Trolox containing 50 mM HEPES (pH 7.5), Mg(OAc)$_2$ 5 mM, KOAc 150 mM, ATP 2 mM, RFC 7 nM) for 15 min and incubation with 6 nM WRN in a 10 mM Tris-HCl (pH 8.0) buffer containing 50 mM NaCl for 3 min. Fork regression reaction was initiated by injecting the standard buffer containing 1 mM ATP-Mg$_2$, unless otherwise indicated, using an automated syringe pump system (Fusion 100, Chemyx). To reduce the photo-bleaching, the imaging buffer also contained an enzymatic oxygen scavenger system (1 mg/ml glucose oxidase, 0.8% glucose, 0.04 mg/ml catalase). As excitation sources, 473-nm (Excelsior-473-50-CDRH, Spectra-Physics), 532-nm (Excelsior-532-50-CDRH, Spectra-Physics), and 640-nm (Excelsior-640c-35, Spectra-Physics) lasers were used. Fluorescence signals were collected through a water-immersion objective (UPlanSApo 603, Olympus), separated using two dichroic mirrors (540dcxt, 635dcxr, and 740dcxr, Chroma) and a mirror (BB01-E02, Thorlabs), and imaged on an EM-CCD camera (Ixon DV897, Andor). Scattered laser light was filtered out using a long-pass filter for 535-nm (LP03-532RU-25, Semrock) and a notch filter for 640-nm (NF03-633 E-25, Semrock). Data acquisition, FRET trace extraction, and data analysis were done as described previously[19,20] using home-made programs written in Visual C++ (Microsoft), IDL (ITT), and MATLAB (MathWorks), respectively.

**Flow cytometry.** Cells were labeled with 10 mM EdU for 30 min before harvesting and were processed using the Click-iT® EdU Flow Cytometry Assay Kit (Thermo Fisher Scientific) according to the manufacturer's instructions. In brief, cells were washed with PBS and then incubated with RNase A (0.1 mg/mL) at 37 °C for 1 h. DNA was stained with 0.05 mg/mL propidium iodide. Flow cytometry was performed on a FACSVerse™ flow cytometer using BD FACSuite™ software (BD Biosciences). Data analysis was performed using the FlowJo software.

**Confocal microscope sample preparation.** Cells plated on LabTek™ chamber slides (Thermo Fisher Scientific) were fixed and stained as described previously[17] with slight modifications. Briefly, the cells were pre-extracted with CSK buffer for 10 min before fixation. Cells were then fixed with 4% paraformaldehyde at RT for 20 min. The fixed cells were incubated with the indicated antibodies diluted in PBS supplemented with 10% fetal bovine serum at 4 °C overnight. After three washes with 0.05% Triton X-100™ in PBS, Alexa Fluor®-conjugated secondary antibodies (Thermo Fisher Scientific) were added and incubated for 30 min. Cells were mounted using ProLong® Gold antifade reagent (Vector Laboratories). Confocal images were acquired with an LSM880 confocal microscope (Carl Zeiss). Image acquisition and analysis were performed with the ZEN2.1 software.

**Analysis of metaphase chromosomes.** Cells were incubated for 4 h with 0.2 µg/mL colcemid and then metaphase cells were harvested by trypsinization. The cells were then swollen in 75 mM KCl for 15 min at 37 °C and fixed with methanol:acetic acid (3:1) twice. Cells were dropped onto glass microscope slides and stained with 5% Giemsa stain. Images were acquired using a fluorescence microscope (BX53; Olympus). At least 35 metaphase cells were taken randomly from each condition.

**Mouse and in vivo micronucleus assay.** $Atad5^{+/m}$ mice, which were described previously[13], and wild-type mice were used for the micronucleus assay. All animal care and experimental procedures were approved by the institutional animal care and use committee at the Ulsan National Institute of Science and Technology (UNIST). We performed an in vivo micronucleus assay following the method previously described[27]. Briefly, 12-week-old male mice were intraperitoneally injected with 0.1 g/kg of HU. Fifty microliters of peripheral blood were collected just before HU injection and 24, 48, and 72 h after HU injection. The blood was quickly transferred into heparin solution in a microtube, fixed with 100% methanol, and stored in a deep freezer for at least 12 h. Samples were then stained with an anti-CD71 antibody and propidium iodide, and then analyzed by flow cytometry. Statistical analysis was performed by two-way analysis of variance using Prism 6 software (GraphPad).

**Reporting Summary.** Further information on research design is available in the Nature Research Reporting Summary linked to this Article.

## Data availability
Data supporting the findings of this manuscript are available from the corresponding authors upon reasonable request. The source data underlying Figs. 1d–f, h–j, 2b–g, i–n, 3a–e, g–j, 4a, c–j, 5b–h, 6b, d–e, g, and Supplementary Figs. 1B, 1D, 1F–H, 2A, B, 2D, 3C–H, 4A, B, 4F–G, 5A, 5C, D are provided as a Source Data file.

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

## Acknowledgements
We thank members in the Center for Genomic Integrity, and Institute for Basic Science for helpful discussions and comments on the manuscript. This research was mainly supported by the Institute for Basic Science (IBS-R022-D1). This work was also partially supported by UNIST research fund (1.180063). This work was also partially supported by a grant from the National Research Foundation of Korea to S. Hohng (2019R1A2C2005209).

## Author contribution
S.H.P., N.K. and K.-y.L. performed most of cell-based experiments. M.W. and D.L. did some cell-based experiments. E.A.L., S.Hwang and J.S.R. helped DNA fiber, iPOND, and chromosome analysis, respectively. I.B.P. and J.H.P. did mouse experiment. J.P. and S.K. purified proteins used in the study. E.S. and S.Hohng did single-molecule experiment. S.H.P. and K.-y.L. designed the experiments and analyzed the data. S.H.P., E.S., S.Hohng and K.-y.L. composed the manuscript with input from all authors. K.-y.L. and K.M. directed the project.

## Competing interests
The authors declare no competing interests.
