## [Peer Review File · Nature Communications]

Reviewers' comments:

Reviewer #1 (Remarks to the Author):

Previously, Park and colleagues have shown that ATAD5 interacts with RFC2-5 to form an RFC-like complex (RLC). Unlike RFC, RLC has an opposite role in unloading PCNA from the chromatin. In this study, Park and colleagues described their study on the molecular mechanism underlying RAD51 recruitment to replication fork upon replication stress via the role of ATAD5 in unloading PCNA. They reported that ATAD5 and UAF1 are required for RAD51 recruitment upon replication stress, and that two regions of ATAD5 interact with RAD51. ATAD5 deficiency decreases RPA2 phosphorylation in response to replication stress. Unfortunately, most of the figures are very low resolution and unreadable, making it difficult to interpret the data. In addition, this study did not go far enough to support some of the critical conclusions made by the authors. Some of the observations also contradict with each other without proper discussion (e.g. Figures 5 & 6 show ATAD5 depletion led to both reduced and increasing HU-induced DNA breaks).

Specific comments:

1. Figure 1A: The efficiency of the siRNA knockdown only 14 hrs after transfection needs to be verified by western analysis.
2. Figure 1F: The experiment described in the figure legend does not match the actual figure.
3. Figure 2A: The images are very small and blurred. It appears that there are more green positive cells in the untreated cells (upper panels) than HU treated cells (bottom panels). This doesn't match the quantification shown in Figure 2B. Also, the authors should measure # of cells positive with RAD51 foci, not RAD51 nuclear intensity.
4. Figure 2C-D: Are the western blots on the upper right corner of each of these sub-figures placed there on purpose?
5. There are two figure legends for Figure 2C-D.
6. Figure 2E: The authors should use iPOND to demonstrate the effect of the mutant ATAD5 on Rad51 recruitment. Measuring the levels of the chromatin-bound RAD51 does not provide direct evidence for Rad51 recruitment to the replication fork.
7. Figure 2G: The labels are hard to read, and it is difficult to know what each graph is about.
8. Figure 3D: The 1-700 fragment shows enhanced interaction with RAD51 but not the 1-800 fragment.
9. p.7: The authors need to describe what "small RFCs" are, and what is the difference between small and regular RFCs. Is small RFC = RLC (since it contains ATAD5)?
10. The authors mapped the RAD51-interacting domains of ATAD5, but they did not go further to assess whether these interaction domains are required for RAD51 recruitment to the stalled replication fork. Their interaction may be enhanced by HU, but is the interaction required for the recruitment?
11. Figure 4H & 4J: The function of PCNA is regulated by several post-translational modifications, including ubiquitylation, sumoylation and ISGylation. It has been reported *S. cerevisiae* Elg1 interacts preferentially with SUMOylated PCNA via three SUMO interacting motifs (SIMs) and a PCNA-interacting peptide (PIP)-like motif and the SUMO interaction function of Elg1 and ATAD5 also contributes to genome stability (Nucleic Acids Res. 2017 Apr 7; 45(6): 3189-3203; Genes Dev.

2015 Oct 1;29(19):2067-80). It is not clear how did authors come to the conclusion that the modified PCNA observed in these figures are ubiquitinated PCNA, but not other form(s).

12. It has been shown that PCNA is SUMOylated to recruit PARI to the stalled replication fork and to restrict unscheduled recombination by disrupting the formation of RAD51 nucleofilament. The author needs to exclude the possibility that ATAD5 knockdown in 293T and U2OS can affect PCNA SUMOylation and PARI recruitment upon replication stress to affect RAD51 recruitment to the chromatin.

13. In Figure 5, the authors showed that ATAD5 knockdown decreases HU-induced DSB formation and inhibits MUS81-mediated ssDNA breaks. However, in Figure 6, the authors showed the opposite result that ATAD5 deficiency increases DNA breaks. The authors need to provide a rationale for these two contradictory observations.

14. Figure 6H(i): It is not obvious how a DSB at the regressed fork (as drawn) leads to fork restart. The kind of DNA break shown would be expected to be detrimental to replication fork restart.

Reviewer #2 (Remarks to the Author):

General comments:

This study by Park et al uncovers novel aspects of stalled replication fork processing. ATAD5 clearly has a role in fork restart and fork slowing likely via promoting RAD51 actions, a key factor in the biology of stalled forks. Authors functionally investigate this uncovering that UAF1 interaction-defective and ATPase-defective ATAD5 mutants are important for RAD51 accumulation at the stalled forks. They also uncover that ATAD5 promotes DSBs after more extensive fork stalls, these are mediated by the MUS81 nuclease. Finally, they find that ATAD5 is important for maintaining genome stability.

Opinion:

The study is timely because the biology and mechanisms of stalled fork responses are not fully understood, here, especially links with replisome components such as roles of PCNA. Furthermore, ATAD5 is an emerging potential tumor suppressor which further merits the present study. The authors mainly use standard methods, however, they do utilize a quite elegant FRET method to study fork regression. Overall, the data are of reasonable quality utilizing several approaches and systems to test and validate their points. The data generally support the conclusions in the text. A few aspects are rather weak and should be addressed, which includes the fact that ATAD5 has only moderate impact on stalled fork processes.

Major points:

*Fork restart impact is rather weak compared to other studies (such as Dungrawala et al, 2017), suggesting this is a minor pathway. The authors should address or comment on this. Further epistasis analysis is needed, as example, the authors need to co-deplete ATAD5 and RAD51 to test if they they are on the same pathway.

*Quality of some of the immunoblots makes it challenging to interpret the data. Occasionally, there are issues with inadequate depletions (for instance figure 4F, 2C).

*There is a potential discrepancy between the data in figures 1F and 2F. Authors seems to ignore the increased track length in figure 2F. Statistics is missing from the fiber experiments. The authors should address or comment on this.

Minor points:

- * Figure 2A: There seems to be more positive cells in the non-treated samples. This should be repeated and tested. PCNA should be included in the IPOND experiment.
- * Figure 2C: Due to low quality (could be digital version) it is hard to see it if there is an ATAD5 depletion. This should be repeated and tested.
- * Figures 2K. The differences appear minor (<10%), and authors should consider the effects of RAD51 on reversal.
- * Figure 4E: PCNA not present in control lane, this should be explained?
- * Figure 4F: Total RPA2, MCM and DNA-PK loading missing. ATAD5 depletion is hard to see.
- * Figure 4G: RAD51 recruitment is not impaired in response to siATAD5 after 2h of HU, is this correct?
- * Figure 4I: It seems there is decrease of ATAD5 expression in RAD51 depleted or BO2 treated cells. This should be explained?

Reviewer #3 (Remarks to the Author):

This paper describes very interesting work on the multiple roles of ATAD5 in regulating genome stability and its possible relevance to some cancers. The paper represents a great deal of work, much of it new and some relying on the results of previous reported studies from the same group. The claim of the paper is that ATAD5 regulates the recovery of stalled replication forks through its multiple functions that include the unloading of the PCNA processivity clamp, the recruitment of RAD51 to stalled replication forks, the enhancement of RAD51 under replication stress, and the promotion of ssDNA breaks mediated by MUS81 during the process of DNA fork regression and stalled replication fork recovery.

Although these studies appear to be very interesting, I don't think that the general readership of Nature Communications would be an appropriate audience for this work. I found the presentation of the paper to be overly crowded and difficult to read. The main text of the manuscript refers to six different figures, each of which is composed of (on average) a dozen different panels. In many cases, each panel is itself composed of multiple sub-panels, which are usually too dense to read. The situation is quite similar in the Supporting Information (SI) section. The organization of the manuscript is difficult to follow, as the narrative constantly switches between SI section and main text without a clear delineation between the relative importance of SI versus main text material. In addition, the organization of the figures is often unclear, with the presentation of some results not fitting naturally within the context of other results presented in the same figure.

As an example of my difficulty parsing through this material, the discussion of the single molecule data summarized in Fig. 2G – 2J is much too brief to allow the reader to critically assess the validity of the results. As far as I can tell, there is no indication within the text, SI or methods sections of how many single molecule trials were performed in order to construct the results shown in Fig. 2H – 2I, or how the error bars shown in these panels were determined.

I think this work would be much better served, and better appreciated by its intended audience, if it were to be published in a journal such as Biochemistry, JMB or Nucleic Acids Research with less stringent space limitations. Material from the SI could then be included in the text sections of the ms. It is possible that there could be two or three separate papers, which could be published as back-to-back manuscripts. In this format, the discussion of the work could then be carried out in a more methodical manner that would greatly strengthen its overall presentation.

Sep 09, 2019

Response to referees (NCOMMS-19-08710-T)

Reviewer #1 (Remarks to the Author):

Previously, Park and colleagues have shown that ATAD5 interacts with RFC2-5 to form an RFC-like complex (RLC). Unlike RFC, RLC has an opposite role in unloading PCNA from the chromatin. In this study, Park and colleagues described their study on the molecular mechanism underlying RAD51 recruitment to replication fork upon replication stress via the role of ATAD5 in unloading PCNA. They reported that ATAD5 and UAF1 are required for RAD51 recruitment upon replication stress, and that two regions of ATAD5 interact with RAD51. ATAD5 deficiency decreases RPA2 phosphorylation in response to replication stress. Unfortunately, most of the figures are very low resolution and unreadable, making it difficult to interpret the data. In addition, this study did not go far enough to support some of the critical conclusions made by the authors. Some of the observations also contradict with each other without proper discussion (e.g. Figures 5 & 6 show ATAD5 depletion led to both reduced and increasing HU-induced DNA breaks).

- We replaced low resolution figures with high resolution ones (Figure 1G, Figure 2A, Figure 4B, Supplementary Figure 1I, Supplementary Figure 2E, and Supplementary Figure 4E).
- For other comments, we answered in the corresponding specific comments section.

Specific comments:

1. Figure 1A: The efficiency of the siRNA knockdown only 14 hrs after transfection needs to be verified by western analysis.

- We added the result in Supplementary Figure 1D.

2. Figure 1F: The experiment described in the figure legend does not match the actual figure.

➤ Sorry for our mistake. We corrected it.

“U2OS-TetOn-ATAD5 cell lines expressing wild type or two ATAD5 mutants (Δ UAF1, UAF1 interaction defective mutant; mATPase, ATPase domain mutant) were treated with doxycycline and transfected with ATAD5 siRNA under the Noco-APH condition.” was changed to “U2OS cell lines were transfected with ATAD5 siRNA under the Noco-APH condition.”

3. Figure 2A: The images are very small and blurred. It appears that there are more green positive cells in the untreated cells (upper panels) than HU treated cells (bottom panels). This doesn't match the quantification shown in Figure 2B. Also, the authors should measure # of cells positive with RAD51 foci, not RAD51 nuclear intensity.

➤ It is due to a resolution issue. We replaced the image with high resolution one.

➤ We measured # of cells positive with RAD51 foci, too as the reviewer suggested. However, under HU-treated conditions, the percentage of cells with RAD51 foci over 10 was nearly same between ATAD5-

depleted and control cells (See the graph). Thus, we concluded that

nuclear intensity provides more relevant data. In addition, each RAD51 nuclear foci showed slight different intensity (See the images below). Thus, we measured whole RAD51 intensity in each nucleus.

4. Figure 2C-D: Are the western blots on the upper right corner of each of these sub-figures placed there on purpose?

- It was just due to space limitations.

5. There are two figure legends for Figure 2C-D.

- Sorry for the confusion. We moved Figure 2D to Supplementary figure 2A and re-wrote the figure legends more clearly.

“(C) HEK293T cells were transfected with ATAD5 siRNA for 48 h. (D) U2OS cells expressing ATAD5AID were pre-treated with auxin. (C, D) After depletion of ATAD5, cell were labeled with 10 μM EdU for 20 min, washed, and treated with 2 mM HU for the indicated times. Samples were processed for the isolation of proteins on nascent DNA (iPOND) assay and immunoblotting. The right panel shows chromatin-bound proteins extracted from a portion of cells in (C and D) and immunoblotted.” was changed to “(C) HEK293T cells were

transfected with ATAD5 siRNA for 48 h. Cells were then processed for the isolation of proteins on nascent DNA (iPOND) assay and immunoblotting. The right panel shows chromatin-bound proteins extracted from a portion of cells in (C) by immunoblot analysis.”

6. Figure 2E: The authors should use iPOND to demonstrate the effect of the mutant ATAD5 on Rad51 recruitment. Measuring the levels of the chromatin-bound RAD51 does not provide direct evidence for Rad51 recruitment to the replication fork.

- We agree with the reviewer. We tried the iPOND assay using U2OS-TetON system for recovery experiment (wild type, UAF1 interaction-defective, PCNA unloading-defective ATAD5). However, results were not consistent. We believe it was due to technical difficulty of iPOND using U2OS cells due to large cell numbers required for iPOND. Therefore, we adopted a new assay termed in situ analysis of protein interactions at DNA replication forks (SIRF) (Roy S et al, JCB, 2018), which requires a small number of cells. We performed the recovery experiments with wild type and mutants ATAD5 using the SIRF assay and found that the RAD51 recruitments was defective in cell expressing mutant ATAD5. We added these results in Figure 2E and Supplementary Figures 2B and 2C.

7. Figure 2H: The labels are hard to read, and it is difficult to know what each graph is about.

- We made them easier to read.

8. Figure 3D: The 1-700 fragment shows enhanced interaction with RAD51 but not the 1-800 fragment.

- It was due to experimental variations. We repeated the same experiments several more times and replaced the figure.

9. p.7: The authors need to describe what “small RFCs” are, and what is the difference between small and regular RFCs. Is small RFC = RLC (since it contains ATAD5)?

- We referred small RFCs as four core subunits, RFC2-5. To make clear in the text, we replaced “small RFCs” with “RFC2-5”.

10. The authors mapped the RAD51-interacting domains of ATAD5, but they did not go further to assess whether these interaction domains are required for RAD51 recruitment to the stalled replication fork. Their interaction may be enhanced by HU, but is the interaction required for the recruitment?

- To address this comment, we performed the SIRF assay using the RAD51 interaction-defective ATAD5 and found that RAD51 recruitment was defective. We added the result in Figure 3I and 3J and explained in the text (Page 13, line 2).

Figure 3I

Figure 3J

11. Figure 4H & 4J: The function of PCNA is regulated by several post-translational modifications, including ubiquitylation, sumoylation and ISGylation. It has been reported *S. cerevisiae* Elg1 interacts preferentially with SUMOylated PCNA via three SUMO interacting motifs (SIMs) and a PCNA-interacting peptide (PIP)-like motif and the SUMO interaction function of Elg1 and ATAD5 also contributes to genome stability (Nucleic Acids Res. 2017 Apr 7;45(6):3189-3203; Genes Dev. 2015 Oct 1;29(19):2067-80). It is not clear how did authors come to the conclusion that the modified PCNA observed in these figures are ubiquitinated PCNA, but not other form(s).

- The results presented was obtained by using anti-ubiquityl PCNA (Lys164) antibody (Cell signaling #13439). Thus, we described the results as ubiquitylated PCNA.

12. It has been shown that PCNA is SUMOylated to recruit PARI to the stalled replication fork and to restrict unscheduled recombination by disrupting the formation of RAD51 nucleofilament. The author needs to exclude the possibility that ATAD5 knockdown in 293T and U2OS can affect PCNA SUMOylation and PARI recruitment upon replication stress to affect RAD51 recruitment to the chromatin.

- SUMOylated PCNA is hard to detect in human cells. The papers that the reviewer mentioned also did not provide the endogenous “SUMOylated PCNA” data. Thus, we examined the effects of ATAD5 depletion on PARI recruitment as the reviewer suggested. We found that HU treatment reduced PARI signal, which was shown in both ATAD-depleted and control cells. Although we could not completely exclude SUMOylated PCNA effect in human cells due to detection difficulty, we concluded that at least PARI recruitment to stalled forks was not happened in human cells.

13. In Figure 5, the authors showed that ATAD5 knockdown decreases HU-induced DSB formation and inhibits MUS81-mediated ssDNA breaks. However, in Figure 6, the authors showed the opposite result that ATAD5 deficiency increases DNA breaks. The authors need to provide a rationale for these two contradictory observations.

14. Figure 6H(i): It is not obvious how a DSB at the regressed fork (as drawn) leads to fork restart. The kind of DNA break shown would be expected to be detrimental to replication fork restart.

- MUS81-mediated break has been reported to facilitate fork restart (Pepe et al., 2014, Cell reports) and we described it in the previous version of the manuscript. However, we agree with the reviewer that it seems not clear how the breaks lead to fork restart. We believe that it is beyond the scope of our manuscript to uncover the mechanisms because our manuscript is focused on understanding the role of ATAD5 in the early stages of stalled replication forks. We simply used MUS81-dependent break as a consequential downstream effects for processing stalled replication forks. We think that if fork regression is defective as shown in ATAD5-depleted cells, it leads to less MUS81-mediated break as shown in Figure 5 and inefficient fork restart, which eventually causes genomic instability as marked by chromosome breaks on mitotic spread in Figure 6. We believe there are two types of breaks; one for fork restart mediated by MUS81 which we measured right after 6 h of HU treatment and the other caused by the failure of preserving stalled forks, which is measured after an additional 4 h incubation and detrimental for cells. The former breaks were reduced due to no regression of the fork by ATAD5 depletion. However, due to failure of fork restart, ATAD5 depletion eventually increases DSBs, which results in genomic instability and even cell death.

Reviewer #2 (Remarks to the Author):

General comments:

This study by Park et al uncovers novel aspects of stalled replication fork processing. ATAD5 clearly has a role in fork restart and fork slowing likely via promoting RAD51 actions, a key factor in the biology of stalled forks. Authors functionally investigate this uncovering that UAF1 interaction-defective and ATPase-defective ATAD5 mutants are important for RAD51 accumulation at the stalled forks. They also uncover that ATAD5 promotes DSBs after more extensive fork stalls, these are mediated by the MUS81 nuclease. Finally, they find that ATAD5 is important for maintaining genome stability.

Opinion:

The study is timely because the biology and mechanisms of stalled fork responses are not fully understood, here, especially links with replisome components such as roles of PCNA. Furthermore, ATAD5 is an emerging potential tumor suppressor which further merits the present study. The authors mainly use standard methods, however, they do utilize a quite elegant FRET method to study fork regression. Overall, the data are of reasonable quality utilizing several approaches and systems to test and validate their points. The data generally support the conclusions in the text. A few aspects are rather weak and should be addressed, which includes the fact that ATAD5 has only moderate impact on stalled fork processes.

Major points:

*Fork restart impact is rather weak compared to other studies (such as Dungrawala et al, 2017), suggesting this is a minor pathway. The authors should address or comment on this. Further epistasis analysis is needed, as example, the authors need to co-deplete ATAD5 and RAD51 to test if they are on the same pathway.

- We agree with the reviewer. As the reviewer suggested, we did co-depletion experiments and added the data in the Figure 2D. Co-depletion did not show synergistic or additive effects on stalled fork. Thus, we concluded that ATAD5 and RAD51 regulate fork restart on the same pathway. However, as the reviewer pointed, ATAD5 depletion was less defective than RAD51 depletion. Thus, we added this description and the possibility of a minor pathway in the text ("The less effects in fork restart by ATAD5 depletion compared to RAD51 depletion suggest that there might be an additional pathway for RAD51 regulation." Page 9, line 1)

*Quality of some of the immunoblots makes it challenging to interpret the data. Occasionally, there are issues with inadequate depletions (for instance figure 4F, 2C).

- We repeated some experiments and replaced the figure (Figure 2C, Figure 2D (Supplementary Figure 2A in the revised manuscript), Figure 4F, Figure 4I, and Supplementary Figure 4G).

Minor points:

- * Figure 2C, 2D (Supplementary Figure 2A in the revised manuscript): PCNA should be included in the IPOND experiment
- * Figure 2C: Due to low quality (could be digital version) it is hard to see it if there is an ATAD5 depletion. This should be repeated and tested.

- We repeated the experiments in Figure 2C and replaced the figure.

- We added the PCNA blot in Figure 2D (Supplementary Figure 2A in the revised manuscript) and replaced with higher resolution figures.

* Figure 4F: Total RPA2, MCM and DNA-PK loading missing. ATAD5 depletion is hard to see.

- We repeated the experiments with proper loading controls and replaced the figure.

➤ We found several missing issue in the Supplementary Figure 4G, too. Thus, we repeated the experiments and replaced the figure.

*There is a potential discrepancy between the data in figures 1F and 2F (Figure 2G in the revised manuscript). Authors seems to ignore the increased track length in figure 2F. Statistics is missing from the fiber experiments. The authors should address or comment on this.

➤ Sorry for the confusion. It could be due to experimental variations. We added additional statistics in Figure 1F. We repeated Figure 2G experiment and replaced the figure.

Minor points:

* Figure 2A: There seems to be more positive cells in the non-treated samples. This should be repeated and tested.

- It is due to a resolution issue. We replaced it with high resolution image.

* Figures 2K (Figure 2L in the revised manuscript). The differences appear minor (<10%), and authors should consider the effects of RAD51 on reversal.

- We repeated the experiments and replaced the figure. In addition, we tried the same experiments with the original (2 mM HU for 2h) condition and tested other conditions (3 mM HU for 2h and 2 mM HU for 3h). In conclusion, we observed similar effects. Compared to the published data (Petermann et al., 2010, Mol Cell) showing a difference of about 10%, our data consistently showed difference of 7 to 10%.

* Figure 4E: PCNA not present in control lane, this should be explained?

- It is due to a short exposure. We added a long exposure image in the figure.

* Figure 4G: RAD51 recruitment is not impaired in response to siATAD5 after 2h of HU, is this correct?

- Sorry for the confusion. We consistently observed the reduction of RAD51 recruitment (See below) in iPOND-immunoblot as well as newly included SIF data. We used the blot due to quality of images of other proteins.

* Figure 4I: It seems there is decrease of ATAD5 expression in RAD51 depleted or B02 treated cells. This should be explained?

- Sorry for the confusion. It was not observed in all other blots (See below, too). We replaced the figure.

Reviewer #3 (Remarks to the Author):

This paper describes very interesting work on the multiple roles of ATAD5 in regulating genome stability and its possible relevance to some cancers. The paper represents a great deal of work, much of it new and some relying on the results of previous reported studies from the same group. The claim of the paper is that ATAD5 regulates the recovery of stalled replication forks through its multiple functions that include the unloading of the PCNA processivity clamp, the recruitment of RAD51 to stalled replication forks, the enhancement of RAD51 under replication stress, and the promotion of ssDNA breaks mediated by MUS81

during the process of DNA fork regression and stalled replication fork recovery.

Although these studies appear to be very interesting, I don't think that the general readership of Nature Communications would be an appropriate audience for this work. I found the presentation of the paper to be overly crowded and difficult to read. The main text of the manuscript refers to six different figures, each of which is composed of (on average) a dozen different panels. In many cases, each panel is itself composed of multiple sub-panels, which are usually too dense to read. The situation is quite similar in the Supporting Information (SI) section. The organization of the manuscript is difficult to follow, as the narrative constantly switches between SI section and main text without a clear delineation between the relative importance of SI versus main text material. In addition, the organization of the figures is often unclear, with the presentation of some results not fitting naturally within the context of other results presented in the same figure.

As an example of my difficulty parsing through this material, the discussion of the single molecule data summarized in Fig. 2G – 2J is much too brief to allow the reader to critically assess the validity of the results. As far as I can tell, there is no indication within the text, SI or methods sections of how many single molecule trials were performed in order to construct the results shown in Fig. 2H – 2I, or how the error bars shown in these panels were determined.

- Thanks for the comments. We tried to describe the data as detail as possible in the revised manuscript.

I think this work would be much better served, and better appreciated by its intended audience, if it were to be published in a journal such as Biochemistry, JMB or Nucleic Acids Research with less stringent space limitations. Material from the SI could then be included in the text sections of the ms. It is possible that there could be two or three separate papers, which could be published as back-to-back manuscripts. In this format, the discussion of the work could then be carried out in a more methodical manner that would greatly strengthen its overall presentation.

- Thanks for the comments. However, we still want to publish our results in Nature Communication since we believe the journal publishes many similar studies and gives our manuscript more visible to various readers.

Thank you in advance for your consideration of our manuscript.

Sincerely,

Kyungjae (KJ) Myung

REVIEWERS' COMMENTS:

Reviewer #1 (Remarks to the Author):

The authors have sufficient addressed this reviewer's concerns raised during the original review.

Reviewer #2 (Remarks to the Author):

The authors have now addressed a substantial number of points identified in the first manuscript version. The revised manuscript is clearly improved regarding both text and figure contents, this to a level where it will further advance the field. However, the manuscript text is still not completely up to the level needed to convincingly communicate the findings. Many sentences are too complicated and unclear, or, they do not describe experiments and data adequately. As an example, the last sentence of the Abstract claims "..., which explains frequent ATAD5 mutations in human tumors". This is obviously an overstatement that must be moderated, unfortunately, it is just one of many textual issues.

Park et al., NCOMMS-19-08710A

Responses to reviewers' comments

REVIEWERS' COMMENTS:

Reviewer #1 (Remarks to the Author):

The authors have sufficient addressed this reviewer's concerns raised during the original review.

Thanks

Reviewer #2 (Remarks to the Author):

The authors have now addressed a substantial number of points identified in the first manuscript version. The revised manuscript is clearly improved regarding both text and figure contents, this to a level where it will further advance the field. However, the manuscript text is still not completely up to the level needed to convincingly communicate the findings. Many sentences are too complicated and unclear, or, they do not describe experiments and data adequately. As an example, the last sentence of the Abstract claims "..., which explains frequent ATAD5 mutations in human tumors". This is obviously an overstatement that must be moderated, unfortunately, it is just one of many textual issues.

We thoroughly reviewed the manuscript and changed complex or unclear sentences to make it easier to read. We also paid more attention not to exaggerate our findings. All changes are tracked in the resubmitted manuscript.